# Projection Robust Wasserstein Distance and Riemannian Optimization

**Tianyi Lin**$^{\diamond *}$    **Chenyou Fan**$^{\dagger *}$    **Nhat Ho**$^{\ddagger}$    **Marco Cuturi**$^{\triangleleft,\triangleright}$    **Michael I. Jordan**$^{\diamond}$
University of California, Berkeley$^{\diamond}$
The Chinese University of Hong Kong, Shenzhen$^{\dagger}$
University of Texas, Austin$^{\ddagger}$
CREST - ENSAE$^{\triangleleft}$, Google Brain$^{\triangleright}$
{darren_lin,jordan}@cs.berkeley.edu, fanchenyou@cuhk.edu.cn, minhnhat@utexas.edu
cuturi@google.com

## Abstract

Projection robust Wasserstein (PRW) distance, or Wasserstein projection pursuit (WPP), is a robust variant of the Wasserstein distance. Recent work suggests that this quantity is more robust than the standard Wasserstein distance, in particular when comparing probability measures in high-dimensions. However, it is ruled out for practical application because the optimization model is essentially non-convex and non-smooth which makes the computation intractable. Our contribution in this paper is to revisit the original motivation behind WPP/PRW, but take the hard route of showing that, despite its non-convexity and lack of nonsmoothness, and even despite some hardness results proved by Niles-Weed and Rigollet [68] in a minimax sense, the original formulation for PRW/WPP *can* be efficiently computed in practice using Riemannian optimization, yielding in relevant cases better behavior than its convex relaxation. More specifically, we provide three simple algorithms with solid theoretical guarantee on their complexity bound (one in the appendix), and demonstrate their effectiveness and efficiency by conducing extensive experiments on synthetic and real data. This paper provides a first step into a computational theory of the PRW distance and provides the links between optimal transport and Riemannian optimization.

## 1 Introduction

Optimal transport (OT) theory [86, 87] has become an important source of ideas and algorithmic tools in machine learning and related fields. Examples include contributions to generative modelling [4, 74, 38, 83, 39], domain adaptation [21], clustering [80, 44], dictionary learning [73, 76], text mining [58], neuroimaging [48] and single-cell genomics [75, 91]. The Wasserstein geometry has also provided a simple and useful analytical tool to study latent mixture models [43], reinforcement learning [6], sampling [20, 25, 63, 8] and stochastic optimization [66]. For an overview of OT theory and the relevant applications, we refer to the recent survey [70].

**Curse of Dimensionality in OT.** A significant barrier to the direct application of OT in machine learning lies in some inherent statistical limitations. It is well known that the sample complexity of approximating Wasserstein distances between densities using only samples can grow exponentially in dimension [29, 36, 89, 53]. Practitioners have long been aware of this issue of the curse of dimensionality in applications of OT, and it can be argued that most of the efficient computational schemes that are known to improve computational complexity also carry out, implicitly through their simplifications, some form of statistical regularization. There have been many attempts to

---

$^{*}$Tianyi Lin and Chenyou Fan contributed equally to this work.

mitigate this curse when using OT, whether through entropic regularization [24, 23, 40, 61]; other regularizations [28, 11]; quantization [17, 35]; simplification of the dual problem in the case of 1-Wasserstein distance [78, 4] or by only using second-order moments of measures to fall back on the Bures-Wasserstein distance [9, 65, 19].

**Subspace projections: PRW and WPP.** We focus in this paper on another important approach to regularize the Wasserstein distance: Project input measures onto lower-dimensional subspaces and compute the Wasserstein distance between these reductions, instead of the original measures. The simplest and most representative example of this approach is the sliced Wasserstein distance [71, 14, 52, 67], which is defined as the average Wasserstein distance obtained between random 1D projections. In an important extension, Paty and Cuturi [69] and Niles-Weed and Rigollet [68] proposed very recently to look for the $k$-dimensional subspace ($k > 1$) that would maximize the Wasserstein distance between two measures after projection. [69] called that quantity the *projection robust Wasserstein* (PRW) distance, while [68] named it *Wasserstein Projection Pursuit* (WPP). PRW/WPP are conceptually simple, easy to interpret, and do solve the curse of dimensionality in the so called spiked model as proved in [68, Theorem 1] by recovering an optimal $1/\sqrt{n}$ rate. Very recently, Lin et al. [59] further provided several fundamental statistical bounds for PRW as well as asymptotic guarantees for learning generative models with PRW. Despite this appeal, [69] quickly rule out PRW for practical applications because it is non-convex, and fall back on a convex relaxation, called the *subspace robust Wasserstein* (SRW) distance, which is shown to work better empirically than the usual Wasserstein distance. Similarly, [68] seem to lose hope that it can be computed, by stating *"it is unclear how to implement WPP efficiently,"* and after having proved positive results on sample complexity, conclude their paper on a negative note, showing hardness results which apply for WPP when the ground cost is the Euclidean metric (the 1-Wasserstein case). Our contribution in this paper is to revisit the original motivation behind WPP/PRW, but take the hard route of showing that, despite its non-convexity and lack of nonsmoothness, and even despite some hardness results proved in [68] in a minimax sense, the original formulation for PRW/WPP *can* be efficiently computed in practice using Riemannian optimization, yielding in relevant cases better behavior than SRW. For simplicity, we refer from now on to PRW/WPP as PRW.

**Contribution:** In this paper, we study the computation of the PRW distance between two discrete probability measures of size $n$. We show that the resulting optimization problem has a special structure, allowing it to be solved in an efficient manner using Riemannian optimization [2, 16, 50, 18]. Our contributions can be summarized as follows.

1. We propose a max-min optimization model for computing the PRW distance. The maximization and minimization are performed over the Stiefel manifold and the transportation polytope, respectively. We prove the existence of the subdifferential (Lemma 2.2), which allows us to properly define an $\epsilon$-*approximate pair of optimal subspace projection and optimal transportation plan* (Definition 2.7) and carry out a finite-time analysis of the algorithm.

2. We define an entropic regularized PRW distance between two finite discrete probability measures, and show that it is possible to efficiently optimize this distance over the transportation polytope using the Sinkhorn iteration. This poses the problem of performing the maximization over the Stiefel manifold, which is not solvable by existing optimal transport algorithms [24, 3, 30, 56, 57, 42]. To this end, we propose two new algorithms, which we refer to as *Riemannian gradient ascent with Sinkhorn* (RGAS) and *Riemannian adaptive gradient ascent with Sinkhorn* (RAGAS), for computing the entropic regularized PRW distance. These two algorithms are guaranteed to return an $\epsilon$-*approximate pair of optimal subspace projection and optimal transportation plan* with a complexity bound of $\widetilde{O}(n^2 d \|C\|_\infty^4 \epsilon^{-4} + n^2 \|C\|_\infty^8 \epsilon^{-8} + n^2 \|C\|_\infty^{12} \epsilon^{-12})$. To the best of our knowledge, our algorithms are the first provably efficient algorithms for the computation of the PRW distance.

3. We provide comprehensive empirical studies to evaluate our algorithms on synthetic and real datasets. Experimental results confirm our conjecture that the PRW distance performs better than its convex relaxation counterpart, the SRW distance. Moreover, we show that the RGAS and RAGAS algorithms are faster than the Frank-Wolfe algorithm while the RAGAS algorithm is more robust than the RGAS algorithm.

**Organization.** The remainder of the paper is organized as follows. In Section 2, we present the nonconvex max-min optimization model for computing the PRW distance and its entropic regularized version. We also briefly summarize various concepts of geometry and optimization over the Stiefel manifold. In Section 3, we propose and analyze the RGAS and RAGAS algorithms for computing the

entropic regularized PRW distance and prove that both algorithms achieve the finite-time guarantee under stationarity measure. In Section 4, we conduct extensive experiments on both synthetic and real datasets, demonstrating that the PRW distance provides a computational advantage over the SRW distance in real application problems. In the supplementary material, we provide further background materials on Riemannian optimization, experiments with the algorithms, and proofs for key results. For the sake of completeness, we derive a near-optimality condition (Definition E.1 and E.2) for the max-min optimization model and propose another *Riemannian SuperGradient Ascent with Network simplex iteration* (RSGAN) algorithm for computing the PRW distance without regularization and prove the finite-time convergence under the near-optimality condition.

**Notation.** We let $[n]$ be the set $\{1, 2, \ldots, n\}$. $\mathbf{1}_n$ and $\mathbf{0}_n$ are the $n$-dimension vectors of ones and zeros. $\Delta^n = \{u \in \mathbb{R}^n : \mathbf{1}_n^\top u = 1, u \geq \mathbf{0}_n\}$ is the probability simplex. For $x \in \mathbb{R}^n$ and $p \in (1, +\infty)$, the $\ell_p$-norm stands for $\|x\|_p$ and the Dirac delta function at $x$ stands for $\delta_x(\cdot)$. Diag $(x)$ denotes an $n \times n$ diagonal matrix with $x$ as the diagonal elements. For $X \in \mathbb{R}^{n \times n}$, the right and left marginals are denoted $r(X) = X\mathbf{1}_n$ and $c(X) = X^\top \mathbf{1}_n$, and $\|X\|_\infty = \max_{1 \leq i,j \leq n} |X_{ij}|$ and $\|X\|_1 = \sum_{1 \leq i,j \leq n} |X_{ij}|$. The notation diag$(X)$ stands for an $n$-dimensional vector which corresponds to the diagonal elements of $X$. If $X$ is symmetric, $\lambda_{\max}(X)$ stands for its largest eigenvalue. $\text{St}(d, k) := \{X \in \mathbb{R}^{d \times k} : X^\top X = I_k\}$ denotes the Stiefel manifold. For $X, Y \in \mathbb{R}^{n \times n}$, $\langle X, Y \rangle = \text{Trace}(X^\top Y)$ denotes the Euclidean inner product and $\|X\|_F$ denotes the Frobenius norm of $X$. We let $P_{\mathcal{S}}$ be the orthogonal projection onto a closed set $\mathcal{S}$ and $\text{dist}(X, \mathcal{S}) = \inf_{Y \in \mathcal{S}} \|X - Y\|_F$ denotes the distance between $X$ and $\mathcal{S}$. Lastly, $a = O(b(n, d, \epsilon))$ stands for the upper bound $a \leq C \cdot b(n, d, \epsilon)$ where $C > 0$ is independent of $n$ and $1/\epsilon$ and $a = \widetilde{O}(b(n, d, \epsilon))$ indicates the previous inequality where $C$ depends on the logarithmic factors of $n$, $d$ and $1/\epsilon$.

## 2 Projection Robust Wasserstein Distance

In this section, we present the basic setup and optimality conditions for the computation of the projection robust 2-Wasserstein (PRW) distance between two discrete probability measures with at most $n$ components. We also review basic ideas in Riemannian optimization.

### 2.1 Structured max-min optimization model

In this section we define the PRW distance [69] and show that computing the PRW distance between two discrete probability measures supported on at most $n$ points reduces to solving a structured max-min optimization model over the Stiefel manifold and the transportation polytope.

Let $\mathscr{P}(\mathbb{R}^d)$ be the set of Borel probability measures in $\mathbb{R}^d$ and let $\mathscr{P}_2(\mathbb{R}^d)$ be the subset of $\mathscr{P}(\mathbb{R}^d)$ consisting of probability measures that have finite second moments. Let $\mu, \nu \in \mathscr{P}_2(\mathbb{R}^d)$ and $\Pi(\mu, \nu)$ be the set of couplings between $\mu$ and $\nu$. The 2-Wasserstein distance [87] is defined by

$$\mathcal{W}_2(\mu, \nu) := \left( \inf_{\pi \in \Pi(\mu, \nu)} \int \|x - y\|^2 \, d\pi(x, y) \right)^{1/2}. \tag{2.1}$$

To define the PRW distance, we require the notion of the push-forward of a measure by an operator. Letting $\mathcal{X}, \mathcal{Y} \subseteq \mathbb{R}^d$ and $T : \mathcal{X} \to \mathcal{Y}$, the push-forward of $\mu \in \mathscr{P}(\mathcal{X})$ by $T$ is defined by $T_\# \mu \in \mathscr{P}(\mathcal{Y})$. In other words, $T_\# \mu$ is the measure satisfying $T_\# \mu(A) = \mu(T^{-1}(A))$ for any Borel set in $\mathcal{Y}$.

**Definition 2.1** *For $\mu, \nu \in \mathscr{P}_2(\mathbb{R}^d)$, let $\mathcal{G}_k = \{E \subseteq \mathbb{R}^d \mid \dim(E) = k\}$ be the Grassmannian of $k$-dimensional subspace of $\mathbb{R}^d$ and let $P_E$ be the orthogonal projector onto $E$ for all $E \in \mathcal{G}_k$. The $k$-dimensional PRW distance is defined as $\mathcal{P}_k(\mu, \nu) := \sup_{E \in \mathcal{G}_k} \mathcal{W}_2(P_{E\#}\mu, P_{E\#}\nu)$.*

Paty and Cuturi [69, Proposition 5] have shown that there exists a subspace $E^* \in \mathcal{G}_k$ such that $\mathcal{P}_k(\mu, \nu) = \mathcal{W}_2(P_{E^*\#}\mu, P_{E^*\#}\nu)$ for any $k \in [d]$ and $\mu, \nu \in \mathscr{P}_2(\mathbb{R}^d)$. For any $E \in \mathcal{G}_k$, the mapping $\pi \mapsto \int \|P_E(x - y)\|^2 \, d\pi(x, y)$ is lower semi-continuous. This together with the compactness of $\Pi(\mu, \nu)$ implies that the infimum is a minimum. Therefore, we obtain a structured max-min optimization problem:

$$\mathcal{P}_k(\mu, \nu) = \max_{E \in \mathcal{G}_k} \min_{\pi \in \Pi(\mu, \nu)} \left( \int \|P_E(x - y)\|^2 \, d\pi(x, y) \right)^{1/2}. \tag{2.2}$$

Let us now consider this general problem in the case of discrete probability measures, which is the focus of the current paper. Let $\{x_1, x_2, \ldots, x_n\} \subseteq \mathbb{R}^d$ and $\{y_1, y_2, \ldots, y_n\} \subseteq \mathbb{R}^d$ denote sets

---
**Algorithm 1** Riemannian Gradient Ascent with Sinkhorn Iteration (RGAS)
---
1: **Input:** $\{(x_i, r_i)\}_{i \in [n]}$ and $\{(y_j, c_j)\}_{j \in [n]}$, $k = \widetilde{O}(1)$, $U_0 \in \mathrm{St}(d, k)$ and $\epsilon$.
2: **Initialize:** $\widehat{\epsilon} \leftarrow \frac{\epsilon}{10\|C\|_\infty}$, $\eta \leftarrow \frac{\epsilon \min\{1, 1/\bar{\theta}\}}{40 \log(n)}$ and $\gamma \leftarrow \frac{1}{(8L_1^2 + 16L_2)\|C\|_\infty + 16\eta^{-1} L_1^2 \|C\|_\infty^2}$.
3: **for** $t = 0, 1, 2, \ldots$ **do**
4:    Compute $\pi_{t+1} \leftarrow \mathrm{REGOT}(\{(x_i, r_i)\}_{i \in [n]}, \{(y_j, c_j)\}_{j \in [n]}, U_t, \eta, \widehat{\epsilon})$.
5:    Compute $\xi_{t+1} \leftarrow P_{\mathrm{T}_{U_t}\mathrm{St}}(2V_{\pi_{t+1}}U_t)$.
6:    Compute $U_{t+1} \leftarrow \mathrm{Retr}_{U_t}(\gamma \xi_{t+1})$.
7: **end for**
---

of $n$ atoms, and let $(r_1, r_2, \ldots, r_n) \in \Delta^n$ and $(c_1, c_2, \ldots, c_n) \in \Delta^n$ denote weight vectors. We define discrete probability measures $\mu := \sum_{i=1}^n r_i \delta_{x_i}$ and $\nu := \sum_{j=1}^n c_j \delta_{y_j}$. In this setting, the computation of the $k$-dimensional PRW distance between $\mu$ and $\nu$ reduces to solving a structured max-min optimization model where the maximization and minimization are performed over the Stiefel manifold $\mathrm{St}(d, k) := \{U \in \mathbb{R}^{d \times k} \mid U^\top U = I_k\}$ and the transportation polytope $\Pi(\mu, \nu) := \{\pi \in \mathbb{R}_+^{n \times n} \mid r(\pi) = r, \ c(\pi) = c\}$ respectively. Formally, we have

$$\max_{U \in \mathbb{R}^{d \times k}} \min_{\pi \in \mathbb{R}_+^{n \times n}} \sum_{i=1}^n \sum_{j=1}^n \pi_{i,j} \|U^\top x_i - U^\top y_j\|^2 \quad \text{s.t. } U^\top U = I_k, \ r(\pi) = r, \ c(\pi) = c. \quad (2.3)$$

The computation of this PRW distance raises numerous challenges. Indeed, there is no guarantee for finding a global Nash equilibrium as the special case of nonconvex optimization is already NP-hard [64]; moreover, Sion's minimax theorem [79] is not applicable here due to the lack of quasi-convex-concave structure. More practically, solving Eq. (2.3) is expensive since (i) preserving the orthogonality constraint requires the singular value decompositions (SVDs) of a $d \times d$ matrix, and (ii) projecting onto the transportation polytope results in a costly quadratic network flow problem. To avoid this, [69] proposed a convex surrogate for Eq. (2.3):

$$\max_{0 \preceq \Omega \preceq I_d} \min_{\pi \in \mathbb{R}_+^{n \times n}} \sum_{i=1}^n \sum_{j=1}^n \pi_{i,j} (x_i - y_j)^\top \Omega (x_i - y_j), \quad \text{s.t. } \mathrm{Trace}(\Omega) = k, \ r(\pi) = r, \ c(\pi) = c. \quad (2.4)$$

Eq. (2.4) is intrinsically a bilinear minimax optimization model which makes the computation tractable. Indeed, the constraint set $\mathcal{R} = \{\Omega \in \mathbb{R}^{d \times d} \mid 0 \preceq \Omega \preceq I_d, \mathrm{Trace}(\Omega) = k\}$ is convex and the objective function is bilinear since it can be rewritten as $\langle \Omega, \sum_{i=1}^n \sum_{j=1}^n \pi_{i,j}(x_i - y_j)(x_i - y_j)^\top \rangle$. Eq. (2.4) is, however, only a convex relaxation of Eq. (2.3) and its solutions are not necessarily good approximate solutions for the original problem. Moreover, the existing algorithms for solving Eq. (2.4) are also unsatisfactory—in each loop, we need to solve a OT or entropic regularized OT exactly and project a $d \times d$ matrix onto the set $\mathcal{R}$ using the SVD decomposition, both of which are computationally expensive as $d$ increases (see Algorithm 1 and 2 in [69]).

## 2.2 Entropic regularized projection robust Wasserstein

Eq. (2.3) has special structure: Fixing a $U \in \mathrm{St}(d, k)$, it reduces to minimizing a linear function over the transportation polytope, i.e., the OT problem. Thus, Eq. (2.3) is to maximize the function $f(U) := \min_{\pi \in \Pi(\mu,\nu)} \sum_{i=1}^n \sum_{j=1}^n \pi_{i,j} \|U^\top x_i - U^\top y_j\|^2$ over the Stiefel manifold $\mathrm{St}(d, k)$.

Since the OT problem admits multiple optimal solutions, $f$ is not differentiable which makes the optimization over the Stiefel manifold hard [1]. Computations are greatly facilitated by adding smoothness, which allows the use of gradient-type and adaptive gradient-type algorithms. This inspires us to consider an entropic regularized version of Eq. (2.3), where an entropy penalty is added to the PRW distance. The resulting optimization model is as follows:

$$\max_{U \in \mathbb{R}^{d \times k}} \min_{\pi \in \mathbb{R}_+^{n \times n}} \sum_{i=1}^n \sum_{j=1}^n \pi_{i,j} \|U^\top x_i - U^\top y_j\|^2 - \eta H(\pi) \quad \text{s.t. } U^\top U = I_k, \ r(\pi) = r, \ c(\pi) = c, \quad (2.5)$$

where $\eta > 0$ is the regularization parameter and $H(\pi) := -\langle \pi, \log(\pi) - \mathbf{1}_n \mathbf{1}_n^\top \rangle$ denotes the entropic regularization term. We refer to Eq. (2.5) as the computation of *entropic regularized PRW distance*. Accordingly, we define the function $f_\eta = \min_{\pi \in \Pi(\mu,\nu)}\{\sum_{i=1}^n \sum_{j=1}^n \pi_{i,j} \|U^\top x_i - U^\top y_j\|^2 - \eta H(\pi)\}$ and reformulate Eq. (2.5) as the maximization of the differentiable function $f_\eta$

---

**Algorithm 2** Riemannian Adaptive Gradient Ascent with Sinkhorn Iteration (RAGAS)

1: **Input:** $\{(x_i, r_i)\}_{i \in [n]}$ and $\{(y_j, c_j)\}_{j \in [n]}$, $k = \widetilde{O}(1)$, $U_0 \in \mathrm{St}(d, k)$, $\epsilon$ and $\alpha \in (0, 1)$.
2: **Initialize:** $p_0 = \mathbf{0}_d$, $q_0 = \mathbf{0}_k$, $\widehat{p}_0 = \alpha\|C\|_\infty^2 \mathbf{1}_d$, $\widehat{q}_0 = \alpha\|C\|_\infty^2 \mathbf{1}_k$, $\widehat{\epsilon} \leftarrow \frac{\epsilon\sqrt{\alpha}}{20\|C\|_\infty}$, $\eta \leftarrow \frac{\epsilon \min\{1, 1/\bar{\theta}\}}{40 \log(n)}$ and
   $\gamma \leftarrow \frac{\alpha}{16L_1^2 + 32L_2 + 32\eta^{-1}L_1^2\|C\|_\infty}$.
3: **for** $t = 0, 1, 2, \ldots$ **do**
4:     Compute $\pi_{t+1} \leftarrow \mathrm{REGOT}(\{(x_i, r_i)\}_{i \in [n]}, \{(y_j, c_j)\}_{j \in [n]}, U_t, \eta, \widehat{\epsilon})$.
5:     Compute $G_{t+1} \leftarrow P_{\mathrm{T}_{U_t}\mathrm{St}}(2V_{\pi_{t+1}}U_t)$.
6:     Update $p_{t+1} \leftarrow \beta p_t + (1 - \beta)\mathrm{diag}(G_{t+1}G_{t+1}^\top)/k$ and $\widehat{p}_{t+1} \leftarrow \max\{\widehat{p}_t, p_{t+1}\}$.
7:     Update $q_{t+1} \leftarrow \beta q_t + (1 - \beta)\mathrm{diag}(G_{t+1}^\top G_{t+1})/d$ and $\widehat{q}_{t+1} \leftarrow \max\{\widehat{q}_t, q_{t+1}\}$.
8:     Compute $\xi_{t+1} \leftarrow P_{\mathrm{T}_{U_t}\mathrm{St}}(\mathrm{Diag}\,(\widehat{p}_{t+1})^{-1/4}G_{t+1}\mathrm{Diag}\,(\widehat{q}_{t+1})^{-1/4})$.
9:     Compute $U_{t+1} \leftarrow \mathrm{Retr}_{U_t}(\gamma\xi_{t+1})$.
10: **end for**

---

over the Stiefel manifold $\mathrm{St}(d, k)$. Indeed, for any $U \in \mathrm{St}(d, k)$ and a fixed $\eta > 0$, there exists a unique solution $\pi^* \in \Pi(\mu, \nu)$ such that $\pi \mapsto \sum_{i=1}^n \sum_{j=1}^n \pi_{i,j}\|U^\top x_i - U^\top y_j\|^2 - \eta H(\pi)$ is minimized at $\pi^*$. When $\eta$ is large, the optimal value of Eq. (2.5) may yield a poor approximation of Eq. (2.3). To guarantee a good approximation, we scale the regularization parameter $\eta$ as a function of the desired accuracy of the approximation. Formally, we consider the following relaxed optimality condition for $\widehat{\pi} \in \Pi(\mu, \nu)$ given $U \in \mathrm{St}(d, k)$.

**Definition 2.2** *The transportation plan $\widehat{\pi} \in \Pi(\mu, \nu)$ is called an $\epsilon$-approximate optimal transportation plan for a given $U \in \mathrm{St}(d, k)$ if the following inequality holds:*

$$\sum_{i=1}^n \sum_{j=1}^n \widehat{\pi}_{i,j}\|U^\top x_i - U^\top y_j\|^2 \leq \min_{\pi \in \Pi(\mu, \nu)} \sum_{i=1}^n \sum_{j=1}^n \pi_{i,j}\|U^\top x_i - U^\top y_j\|^2 + \epsilon. \tag{2.6}$$

### 2.3 Optimality condition

Recall that the computation of the PRW distance in Eq. (2.3) and the entropic regularized PRW distance in Eq. (2.5) are equivalent to

$$\max_{U \in \mathrm{St}(d,k)} \left\{ f(U) := \min_{\pi \in \Pi(\mu, \nu)} \sum_{i=1}^n \sum_{j=1}^n \pi_{i,j}\|U^\top x_i - U^\top y_j\|^2 \right\}, \tag{2.7}$$

and

$$\max_{U \in \mathrm{St}(d,k)} \left\{ f_\eta(U) := \min_{\pi \in \Pi(\mu, \nu)} \sum_{i=1}^n \sum_{j=1}^n \pi_{i,j}\|U^\top x_i - U^\top y_j\|^2 - \eta H(\pi) \right\}. \tag{2.8}$$

Since $\mathrm{St}(d, k)$ is a compact matrix submanifold of $\mathbb{R}^{d \times k}$ [15], Eq. (2.7) and Eq. (2.8) are both special instances of the Stiefel manifold optimization problem. The dimension of $\mathrm{St}(d, k)$ is equal to $dk - k(k + 1)/2$ and the tangent space at the point $Z \in \mathrm{St}(d, k)$ is defined by $\mathrm{T}_Z\mathrm{St} := \{\xi \in \mathbb{R}^{d \times k} : \xi^\top Z + Z^\top \xi = 0\}$. We endow $\mathrm{St}(d, k)$ with Riemannian metric inherited from the Euclidean inner product $\langle X, Y \rangle$ for any $X, Y \in \mathrm{T}_Z\mathrm{St}$ and $Z \in \mathrm{St}(d, k)$. Then the projection of $G \in \mathbb{R}^{d \times k}$ onto $\mathrm{T}_Z\mathrm{St}$ is given by Absil et al. [2, Example 3.6.2]: $P_{\mathrm{T}_Z\mathrm{St}}(G) = G - Z(G^\top Z + Z^\top G)/2$. We make use of the notion of a *retraction*, which is the first-order approximation of an exponential mapping on the manifold and which is amenable to computation [2, Definition 4.1.1]. For the Stiefel manifold, we have the following definition:

**Definition 2.3** *A retraction on $\mathrm{St} \equiv \mathrm{St}(d, k)$ is a smooth mapping $\mathrm{Retr} : \mathrm{TSt} \to \mathrm{St}$ from the tangent bundle $\mathrm{TSt}$ onto $\mathrm{St}$ such that the restriction of $\mathrm{Retr}$ onto $\mathrm{T}_Z\mathrm{St}$, denoted by $\mathrm{Retr}_Z$, satisfies that (i) $\mathrm{Retr}_Z(0) = Z$ for all $Z \in \mathrm{St}$ where $0$ denotes the zero element of $\mathrm{TSt}$, and (ii) for any $Z \in \mathrm{St}$, it holds that $\lim_{\xi \in \mathrm{T}_Z\mathrm{St}, \xi \to 0} \|\mathrm{Retr}_Z(\xi) - (Z + \xi)\|_F / \|\xi\|_F = 0$.*

We now present a novel approach to exploiting the structure of $f$. We begin with several definitions.

**Definition 2.4** *The coefficient matrix between $\mu = \sum_{i=1}^n r_i\delta_{x_i}$ and $\nu = \sum_{j=1}^n c_j\delta_{y_j}$ is defined by $C = (C_{ij})_{1 \leq i,j \leq n} \in \mathbb{R}^{n \times n}$ with each entry $C_{ij} = \|x_i - y_j\|^2$.*

**Definition 2.5** *The* correlation matrix *between* $\mu = \sum_{i=1}^{n} r_i \delta_{x_i}$ *and* $\nu = \sum_{j=1}^{n} c_j \delta_{y_j}$ *is defined by* $V_\pi = \sum_{i=1}^{n} \sum_{j=1}^{n} \pi_{i,j} (x_i - y_j)(x_i - y_j)^\top \in \mathbb{R}^{d \times d}$.

**Lemma 2.1** *The function $f$ is $2\|C\|_\infty$-weakly concave.*

**Lemma 2.2** *Each element of the subdifferential $\partial f(U)$ is bounded by $2\|C\|_\infty$ for all $U \in \mathrm{St}(d,k)$.*

**Remark 2.3** *Lemma 2.1 implies there exists a concave function $g : \mathbb{R}^{d \times k} \to \mathbb{R}$ such that $f(U) = g(U) + \|C\|_\infty \|U\|_F^2$ for any $U \in \mathbb{R}^{d \times k}$. Since $g$ is concave, $\partial g$ is well defined and Vial [85, Proposition 4.6] implies that $\partial f(U) = \partial g(U) + 2\|C\|_\infty U$ for all $U \in \mathbb{R}^{d \times k}$.*

This result together with Vial [85, Proposition 4.5] and Yang et al. [92, Theorem 5.1] lead to the Riemannian subdifferential defined by $\mathrm{subdiff}\, f(U) = P_{\mathrm{T}_U \mathrm{St}}(\partial f(U))$ for all $U \in \mathrm{St}(d,k)$.

**Definition 2.6** *The subspace projection $\widehat{U} \in \mathrm{St}(d,k)$ is called an $\epsilon$-approximate optimal subspace projection of $f$ over $\mathrm{St}(d,k)$ in Eq. (2.7) if it satisfies $\mathrm{dist}(0, \mathrm{subdiff}\, f(\widehat{U})) \le \epsilon$.*

**Definition 2.7** *The pair of subspace projection and transportation plan $(\widehat{U}, \widehat{\pi}) \in \mathrm{St}(d,k) \times \Pi(\mu, \nu)$ is an $\epsilon$-approximate pair of optimal subspace projection and optimal transportation plan for the computation of the PRW distance in Eq. (2.3) if the following statements hold true: (i) $\widehat{U}$ is an $\epsilon$-approximate optimal subspace projection of $f$ over $\mathrm{St}(d,k)$ in Eq. (2.7). (ii) $\widehat{\pi}$ is an $\epsilon$-approximate optimal transportation plan for the subspace projection $\widehat{U}$.*

The goal of this paper is to develop a set of algorithms which are guaranteed to converge to a pair of approximate optimal subspace projection and optimal transportation plan, which stand for a stationary point of the max-min optimization model in Eq. (2.3). In the next section, we provide the detailed scheme of our algorithm as well as the finite-time theoretical guarantee.

## 3 Riemannian (Adaptive) Gradient meets Sinkhorn Iteration

We present the *Riemannian gradient ascent with Sinkhorn* (RGAS) algorithm for solving Eq. (2.8). By the definition of $V_\pi$ (cf. Definition 2.5), we can rewrite $f_\eta(U) = \min_{\pi \in \Pi(\mu,\nu)} \{\langle UU^\top, V_\pi \rangle - \eta H(\pi)\}$. Fix $U \in \mathbb{R}^{d \times k}$, and define the mapping $\pi \mapsto \langle UU^\top, V_\pi \rangle - \eta H(\pi)$ with respect to $\ell_1$-norm. By the compactness of the transportation polytope $\Pi(\mu,\nu)$, Danskin's theorem [72] implies that $f_\eta$ is smooth. Moreover, by the symmetry of $V_\pi$, we have

$$\nabla f_\eta(U) = 2V_{\pi^\star(U)}U \quad \text{for any } U \in \mathbb{R}^{d \times k}, \tag{3.1}$$

where $\pi^\star(U) := \mathrm{argmin}_{\pi \in \Pi(\mu,\nu)} \{\langle UU^\top, V_\pi \rangle - \eta H(\pi)\}$. This entropic regularized OT is solved inexactly at each inner loop of the maximization and we use the output $\pi_{t+1} \leftarrow \pi(U_t)$ to obtain an inexact gradient of $f_\eta$ which permits the Riemannian gradient ascent update; see Algorithm 1. Note that the stopping criterion used here is set as $\|\pi_{t+1} - \pi(U_t)\|_1 \le \widehat{\epsilon}$ which implies that $\pi_{t+1}$ is $\epsilon$-approximate optimal transport plan for $U_t \in \mathrm{St}(d,k)$.

The remaining issue is to approximately solve an entropic regularized OT efficiently. We leverage Cuturi's approach and obtain the desired output $\pi_{t+1}$ for $U_t \in \mathrm{St}(d,k)$ using the Sinkhorn iteration. By adapting the proof presented by Dvurechensky et al. [30, Theorem 1], we derive that Sinkhorn iteration achieves a finite-time guarantee which is polynomial in $n$ and $1/\widehat{\epsilon}$. As a practical enhancement, we exploit the matrix structure of $\mathrm{grad}\, f_\eta(U_t)$ via the use of two different adaptive weight vectors, namely $\widehat{p}_t$ and $\widehat{q}_t$; see the adaptive algorithm in Algorithm 2. It is worth mentioning that such an adaptive strategy is proposed by Kasai et al. [50] and has been shown to generate a search direction which is better than the Riemannian gradient $\mathrm{grad}\, f_\eta(U_t)$ in terms of robustness to the stepsize.

**Theorem 3.1** *Either the RGAS algorithm or the RAGAS algorithm returns an $\epsilon$-approximate pair of optimal subspace projection and optimal transportation plan of the computation of the PRW distance in Eq. (2.3) (cf. Definition 2.7) in*

$$\widetilde{O}\left(\left(\frac{n^2 d \|C\|_\infty^2}{\epsilon^2} + \frac{n^2 \|C\|_\infty^6}{\epsilon^6} + \frac{n^2 \|C\|_\infty^{10}}{\epsilon^{10}}\right)\left(1 + \frac{\|C\|_\infty}{\epsilon}\right)^2\right)$$

*arithmetic operations.*

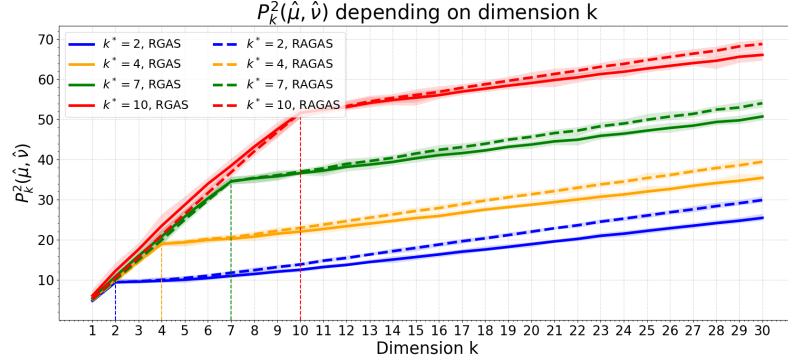

Figure 1: Computation of $\mathcal{P}_k^2(\hat{\mu}, \hat{\nu})$ depending on the dimension $k \in [d]$ and $k^* \in \{2, 4, 7, 10\}$, where $\hat{\mu}$ and $\hat{\nu}$ stand for the empirical measures of $\mu$ and $\nu$ with 100 points. The solid and dash curves are the computation of $\mathcal{P}_k^2(\hat{\mu}, \hat{\nu})$ with the RGAS and RAGAS algorithms, respectively. Each curve is the mean over 100 samples with shaded area covering the min and max values.

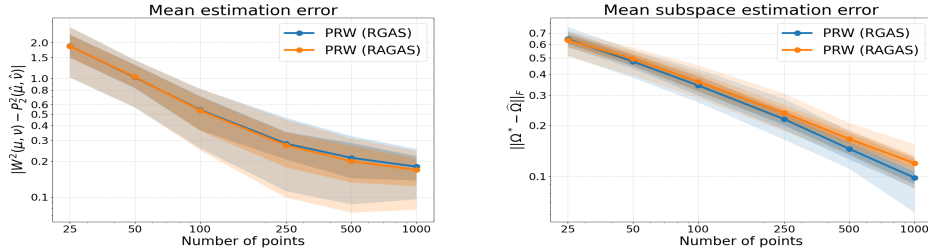

Figure 2: Mean estimation error (left) and mean subspace estimation error (right) over 100 samples, with varying number of points $n$. The shaded areas represent the 10%-90% and 25%-75% quantiles over 100 samples.

**Remark 3.2** *Theorem 3.1 is surprising in that it provides a finite-time guarantee for finding an $\epsilon$-stationary point of a nonsmooth function $f$ over a nonconvex constraint set. This is impossible for general nonconvex nonsmooth optimization even in the Euclidean setting [96, 77]. Our results show that the max-min optimization model in Eq. (2.3) is speical such that fast computation is possible.*

**Remark 3.3** *Note that our algorithms only return an approximate stationary point for the nonconvex max-min optimization model in Eq. (2.3), which needs to be evaluated in practice. It is also interesting to compare such stationary point to the global optimal solution of computing the SRW distance. This is very challenging in general due to multiple stationary points of non-convex max-min optimization model in Eq. (2.3) but possible if the data has certain structure. We leave it to the future work.*

## 4  Experiments

We conduct extensive numerical experiments to evaluate the computation of the PRW distance by the RGAS and RAGAS algorithms. The baseline approaches include the computation of SRW distance with the Frank-Wolfe algorithm[2] [69] and the computation of Wasserstein distance with the POT software package[3] [34]. For the RGAS and RAGAS algorithms, we set $\gamma = 0.01$ unless stated otherwise, $\beta = 0.8$ and $\alpha = 10^{-6}$. The details of our full setup can be found in Appendix G.

**Fragmented hypercube.** Figure 1 presents the behavior of $\mathcal{P}_k^2(\widehat{\mu}, \widehat{\nu})$ as a function of $k^* \in \{2, 4, 7, 10\}$, where $\widehat{\mu}$ and $\widehat{\nu}$ are empirical distributions corresponding to $\mu$ and $\nu$. The sequence is concave and increases slowly after $k = k^*$, which makes sense since the last $d - k^*$ dimensions only represent noise. The rigorous argument for the SRW distance is presented in Paty and Cuturi [69, Proposition 3] but hard to be extended here since the PRW distance is not a sum of eigenvalues. Figure 2 presents mean estimation error and mean subspace estimation error with varying number of points $n \in \{25, 50, 100, 250, 500, 1000\}$. In particular, $\widehat{U}$ is an approximate optimal subspace projection achieved by computing $\mathcal{P}_k^2(\widehat{\mu}, \widehat{\nu})$ with our algorithms and $\Omega^*$ is the optimal projection matrix onto the $k^*$-dimensional subspace spanned by $\{e_j\}_{j \in [k^*]}$. Fixing $k^* = 2$ and construct $\widehat{\mu}$ and

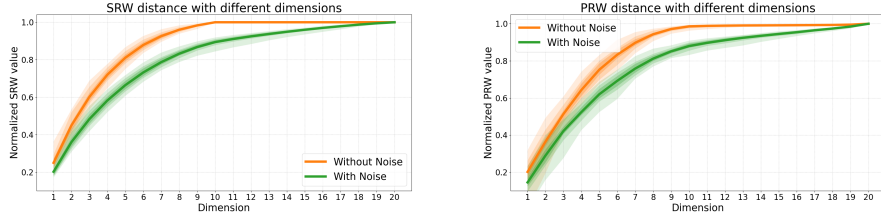

Figure 3: Mean normalized SRW distance (left) and mean normalized PRW distance (right) as a function of dimension. The shaded area shows the 10%-90% and 25%-75% quantiles over the 100 samples.

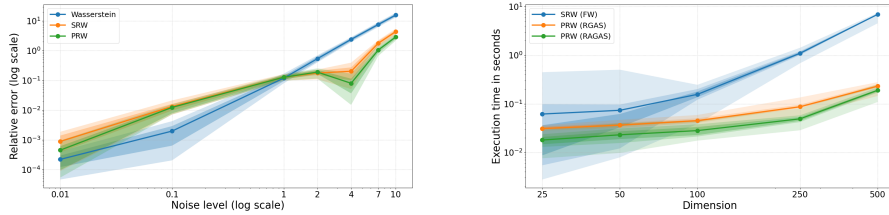

Figure 4: (Left) Comparison of mean relative errors over 100 samples, depending on the noise level. The shaded areas show the min-max values and the 10%-90% quantiles; (Right) Comparisons of mean computation times on CPU (log-log scale). The shaded areas show the minimum and maximum values over the 50 runs.

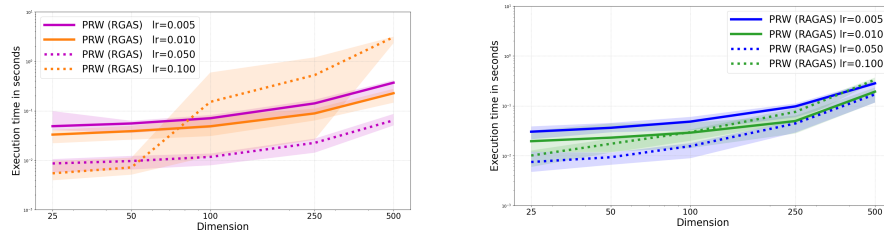

Figure 5: Comparisons of mean computation time of the RGAS and RAGAS algorithms on CPU (log-log scale) for different learning rates. The shaded areas show the max-min values over 50 runs.

$\widehat{\nu}$ from $\mu$ and $\nu$ respectively with $n$ points each, we find that the quality of solutions obtained by the RGAS and RAGAS algorithms are roughly the same.

**Robustness of $\mathcal{P}_k$ to noise.** Figure 3 presents the mean value of $\mathcal{S}_k^2(\widehat{\mu}, \widehat{\nu})/\mathcal{W}_2^2(\widehat{\mu}, \widehat{\nu})$ (left) and $\mathcal{P}_k^2(\widehat{\mu}, \widehat{\nu})/\mathcal{W}_2^2(\widehat{\mu}, \widehat{\nu})$ (right) over 100 samples with varying $k$. We plot the curves for both noise-free and noisy data, where white noise ($\mathcal{N}(0, I_d)$) was added to each data point. With moderate noise, the data is approximately on two 5-dimensional subspaces and both the SRW and PRW distances do not vary too much. Our results are consistent with the SRW distance presented in Paty and Cuturi [69, Figure 6], showing that the PRW distance is also robust to random perturbation of the data. Figure 4 (left) presents the comparison of mean relative errors over 100 samples as the noise level varies. In particular, we construct the empirical measures $\widehat{\mu}_\sigma$ and $\widehat{\nu}_\sigma$ by gradually adding Gaussian noise $\sigma\mathcal{N}(0, I_d)$ to the points. The relative errors of the Wasserstein, SRW and PRW distances are defined the same as in Paty and Cuturi [69, Section 6.3]. For small noise level, the imprecision in the computation of the SRW distance adds to the error caused by the added noise, while the computation of the PRW distance with our algorithms is less sensitive to such noise. When the noise has the moderate to high variance, the PRW distance is the most robust to noise, followed by the SRW distance, both of which outperform the Wasserstein distance.

**Computation time of algorithms.** Considering the fragmented hypercube with dimension $d \in \{25, 50, 100, 250, 500\}$, subspace dimension $k = 2$, number of points $n = 100$ and threshold $\epsilon = 0.001$. For the SRW and the PRW distances, the regularization parameter is set as $\eta = 0.2$ for $n < 250$ and $\eta = 0.5$ otherwise[4], as well as the scaling for the matrix $C$ (cf. Definition 2.4) is applied for stabilizing the algorithms. We stop the RGAS and RAGAS algorithms when $\|U_{t+1} - U_t\|_F/\|U_t\|_F \le \epsilon$. Figure 4 (right) presents the mean computation time of the SRW distance with the Frank-Wolfe algorithm [69] and the PRW distance with our RGAS and RAGAS algorithms. Our

| | D | G | I | KB1 | KB2 | TM | T |
|---|---|---|---|---|---|---|---|
| D | 0/0 | 0.184/0.126 | 0.185/0.135 | 0.195/0.153 | 0.202/0.162 | 0.186/0.134 | **0.170/0.105** |
| G | 0.184/0.126 | 0/0 | **0.172**/0.101 | 0.196/0.146 | 0.203/0.158 | 0.175/**0.095** | 0.184/0.128 |
| I | 0.185/0.135 | 0.172/0.101 | 0/0 | 0.195/0.155 | 0.203/0.166 | **0.169/0.099** | 0.180/0.134 |
| KB1 | 0.195/0.153 | 0.196/0.146 | 0.195/0.155 | 0/0 | **0.164/0.089** | 0.190/0.146 | 0.179/0.132 |
| KB2 | 0.202/0.162 | 0.203/0.158 | 0.203/0.166 | **0.164/0.089** | 0/0 | 0.193/0.155 | 0.180/0.138 |
| TM | 0.186/0.134 | 0.175/**0.095** | **0.169**/0.099 | 0.190/0.146 | 0.193/0.155 | 0/0 | 0.182/0.136 |
| T | **0.170/0.105** | 0.184/0.128 | 0.180/0.134 | 0.179/0.132 | 0.180/0.138 | 0.182/0.136 | 0/0 |

Table 1: Each entry is $\mathcal{S}_k^2/\mathcal{P}_k^2$ distance between different movie scripts. D = Dunkirk, G = Gravity, I = Interstellar, KB1 = Kill Bill Vol.1, KB2 = Kill Bill Vol.2, TM = The Martian, T = Titanic.

| | H5 | H | JC | TMV | O | RJ |
|---|---|---|---|---|---|---|
| H5 | 0/0 | **0.222/0.155** | 0.230/0.163 | 0.228/0.166 | 0.227/0.170 | 0.311/0.272 |
| H | 0.222/0.155 | 0/0 | 0.224/0.163 | 0.221/0.159 | **0.220/0.153** | 0.323/0.264 |
| JC | 0.230/0.163 | 0.224/0.163 | 0/0 | 0.221/**0.156** | **0.219**/0.157 | 0.246/0.191 |
| TMV | 0.228/0.166 | 0.221/0.159 | **0.221**/0.156 | 0/0 | 0.222/**0.154** | 0.292/0.230 |
| O | 0.227/0.170 | 0.220/**0.153** | **0.219**/0.157 | 0.222/0.154 | 0/0 | 0.264/0.215 |
| RJ | 0.311/0.272 | 0.323/0.264 | **0.246/0.191** | 0.292/0.230 | 0.264/0.215 | 0/0 |

Table 2: Each entry is $\mathcal{S}_k^2/\mathcal{P}_k^2$ distance between different Shakespeare plays. H5 = Henry V, H = Hamlet, JC = Julius Caesar, TMV = The Merchant of Venice, O = Othello, RJ = Romeo and Juliet.

approach is significantly faster since the complexity bound of their approach is quadratic in dimension $d$ while our methods are linear in dimension $d$.

**Robustness of algorithms to learning rate.** We use the same experimental setting to evaluate the robustness of our RGAS and RAGAGS algorithms by choosing the learning rate $\gamma \in \{0.01, 0.1\}$. Figure 6 indicates that the RAGAS algorithm is more robust than the RGAS algorithm as the learning rates varies, with smaller variance in computation time (in seconds). This is the case especially when the dimension is large, demonstrating the advantage of the adaptive strategies in practice. To demonstrate the advantage of the adaptive strategies in practice, we initialize the learning rate using four options $\gamma \in \{0.005, 0.01, 0.05, 0.1\}$ and present the results for the RGAS and RAGAS algorithms separately in Figure 5. This is consistent with the results in Figure 6 and supports that the RAGAS algorithm is more robust than the RGAS algorithm to the learning rate.

**Experiments on real data.** We compute the PRW and SRW distances between all pairs of movies in a corpus of seven *movie scripts* and operas in a corpus of eight Shakespeare operas. Each script is tokenized to a list of words, which is transformed to a measure over $\mathbb{R}^{300}$ using WORD2VEC [62] where each weight is word frequency. The SRW and PRW distances between all pairs of movies and operas are in Table 1 and 2, which is consistent with the SRW distance in [69, Figure 9] and shows that the PRW distance is consistently smaller than SRW distance. Additional results on MNIST dataset are deferred to Appendix H.

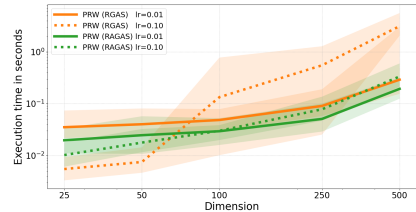

Figure 6: Comparisons of mean computation time of the RGAS and RAGAS algorithms on CPU (log-log scale) for different learning rates. The shaded areas show the max-min values over 50 runs.

**Summary.** The PRW distance has less discriminative power than the SRW distance which is equivalent to the Wasserstein distance [69, Proposition 2]. Such equivalence implies that the SRW distance suffers from the curse of dimensionality in theory. In contrast, the PRW distance has much better sample complexity than the SRW distance if the distributions satisfy the mild condition [68, 59]. Our empirical evaluation shows that the PRW distance is computationally favorable and more robust than the SRW and Wasserstein distance, when the noise has the moderate to high variance.

## 5 Conclusion

We study in this paper the computation of the projection robust Wasserstein (PRW) distance in the discrete setting. A set of algorithms are developed for computing the entropic regularized PRW distance and both guaranteed to converge to an approximate pair of optimal subspace projection and optimal transportation plan. Experiments on synthetic and real datasets demonstrate that our approach to computing the PRW distance is an improvement over existing approaches based on the convex relaxation of the PRW distance and the Frank-Wolfe algorithm. Future work includes the theory for continuous distributions and applications of PRW distance to deep generative models.

## Broader Impact

The paper proposes efficient algorithms with theoretical guarantees to compute distances between probability measures in very high dimensional spaces. The problem of comparing high dimensional probability measures has appeared in several applications in machine learning, statistics, genomics, and neuroscience. Our study, therefore, provides an efficient method for scientists in these domains to deal with large-scale and high dimensional data. We believe that our work is fundamental and initiates new directions in computing high-dimensional probability measures, which leads to faster scientific discoveries. Finally, we do not foresee any negative impact to society from our work.

## 6 Acknowledgements

We would like to thank the area chair and four anonymous referees for constructive suggestions that improve the paper. This work is supported in part by the Mathematical Data Science program of the Office of Naval Research under grant number N00014-18-1-2764.

## Footnotes

[2] Available in https://github.com/francoispierrepaty/SubspaceRobustWasserstein.

[3] Available in https://github.com/PythonOT/POT

[4]Available in https://github.com/francoispierrepaty/SubspaceRobustWasserstein

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
