[Supplementary Material]

# A   Retraction on the Stiefel manifold

The retraction on the Stiefel manifold has the following well-known properties [16, 60] which are important to subsequent analysis in this paper.

**Proposition A.1** *For all $Z \in \text{St} \equiv \text{St}(d, k)$ and $\xi \in \text{T}_Z\text{St}$, there exist constants $L_1 > 0$ and $L_2 > 0$ such that the following two inequalities hold:*

$$\|\text{Retr}_Z(\xi) - Z\|_F \leq L_1\|\xi\|_F,$$
$$\|\text{Retr}_Z(\xi) - (Z + \xi)\|_F \leq L_2\|\xi\|_F^2.$$

For the sake of completeness, we provide four popular restrictions [31, 90, 60, 18] on the Stiefel manifold in practice. Determining which one is the most efficient in the algorithm is still an open question; see the discussion after Liu et al. [60, Theorem 3] and before Chen et al. [18, Fact 3.6].

- **Exponential mapping.** It takes $8dk^2 + O(k^3)$ flops and has the closed-form expression:

$$\text{Retr}_Z^{\exp}(\xi) = [Z \quad Q] \exp\left(\begin{bmatrix} -Z^\top\xi & -R^\top \\ R & 0 \end{bmatrix}\right)\begin{bmatrix} I_k \\ 0 \end{bmatrix}.$$

  where $QR = -(I_k - ZZ^\top)\xi$ is the unique QR factorization.
- **Polar decomposition.** It takes $3dk^2 + O(k^3)$ flops and has the closed-form expression:

$$\text{Retr}_Z^{\text{polar}}(\xi) = (Z + \xi)(I_k + \xi^\top\xi)^{-1/2}.$$

- **QR decomposition.** It takes $2dk^2 + O(k^3)$ flops and has the closed-form expression:

$$\text{Retr}_Z^{\text{qr}}(\xi) = \text{qr}(Z + \xi),$$

  where $\text{qr}(A)$ is the Q factor of the QR factorization of $A$.
- **Cayley transformation.** It takes $7dk^2 + O(k^3)$ flops and has the closed-form expression:

$$\text{Retr}_Z^{\text{cayley}}(\xi) = \left(I_n - \frac{1}{2}W(\xi)\right)^{-1}\left(I_n + \frac{1}{2}W(\xi)\right)Z,$$

  where $W(\xi) = (I_n - ZZ^\top/2)\xi Z^\top - Z\xi^\top(I_n - ZZ^\top/2)$.

# B   Further Technical Lemmas

We first show that $f_\eta$ is continuously differentiable over $\mathbb{R}^{d\times k}$ and the classical gradient inequality holds true over $\text{St}(d, k)$. The derivation is novel and uncovers the structure of the computation of entropic regularized PRW in Eq. (2.5). Let $g : \mathbb{R}^{d\times k} \times \Pi(\mu, \nu) \to \mathbb{R}$ be defined by

$$g(U, \pi) := \sum_{i=1}^n \sum_{j=1}^n \pi_{i,j}\|U^\top x_i - U^\top y_j\|^2 - \eta H(\pi).$$

**Lemma B.1** *$f_\eta$ is differentiable over $\mathbb{R}^{d\times k}$ and $\|\nabla f_\eta(U)\|_F \leq 2\|C\|_\infty$ for all $U \in \text{St}(d, k)$.*

**Lemma B.2** *For all $U_1, U_2 \in \text{St}(d, k)$, the following statement holds true,*

$$|f_\eta(U_1) - f_\eta(U_2) - \langle\nabla f_\eta(U_2), U_1 - U_2\rangle| \leq \left(\|C\|_\infty + \frac{2\|C\|_\infty^2}{\eta}\right)\|U_1 - U_2\|_F^2.$$

**Remark B.3** *Lemma B.2 shows that $f_\eta$ satisfies the classical gradient inequality over the Stiefel manifold. This is indeed stronger than the following statement,*

$$\|\nabla f_\eta(U_1) - \nabla f_\eta(U_2)\|_F \leq \left(2\|C\|_\infty + \frac{4\|C\|_\infty^2}{\eta}\right)\|U_1 - U_2\|_F, \quad \text{for all } U_1, U_2 \in \text{St}(d, k),$$

*and forms the basis for analyzing the complexity bound of Algorithm 1 and 2. The techniques used in proving Lemma B.2 are new and may be applicable to analyze the structure of the robust variant of the Wasserstein distance with other type of regularization [28, 11].*

Before proceeding to the main results, we present a technical lemma on the Hoffman's bound [45, 54] and the characterization of the Hoffman constant [41, 51, 88].

**Lemma B.4** *Consider a polyhedron set* $\mathcal{S} = \{x \in \mathbb{R}^d \mid Ex = t, x \geq 0\}$. *For any point* $x \in \mathbb{R}^d$, *we have*

$$\|x - \text{proj}_{\mathcal{S}}(x)\|_1 \leq \theta(E) \left\| \begin{bmatrix} \max\{0, -x\} \\ Ex - t \end{bmatrix} \right\|_1,$$

*where* $\theta(E)$ *is the Hoffman constant and can be represented by ()*

$$\theta(E) = \sup_{u,v \in \mathbb{R}^d} \left\{ \left\| \begin{bmatrix} u \\ v \end{bmatrix} \right\|_\infty \,\middle|\, \begin{array}{l} \|E^\top v - u\|_\infty = 1, u \geq 0 \\ \text{The corresponding rows of } E \text{ to } v\text{'s nonzero} \\ \text{elements are linearly independent.} \end{array} \right\}$$

We quantify the progress of RGAS algorithm (cf. Algorithm 1) using $f_\eta$ as a potential function and then provide an upper bound for the number of iterations to return an $\epsilon$-approximate optimal subspace projection $U_t \in \text{St}(d, k)$ satisfying $\text{dist}(0, \text{subdiff } f(U_t)) \leq \epsilon$ in Algorithm 1.

**Lemma B.5** *Let* $\{(U_t, \pi_t)\}_{t \geq 1}$ *be the iterates generated by Algorithm 1. We have*

$$\frac{1}{T} \left( \sum_{t=0}^{T-1} \|\text{grad } f_\eta(U_t)\|_F^2 \right) \leq \frac{4\Delta_f}{\gamma T} + \frac{\epsilon^2}{5},$$

*where* $\Delta_f = \max_{U \in \text{St}(d,k)} f_\eta(U) - f_\eta(U_0)$ *is the initial objective gap.*

**Theorem B.6** *Letting* $\{(U_t, \pi_t)\}_{t \geq 1}$ *be the iterates generated by Algorithm 1, the number of iterations required to reach* $\text{dist}(0, \text{subdiff } f(U_t)) \leq \epsilon$ *satisfies that*

$$t = \widetilde{O} \left( \frac{k\|C\|_\infty^2}{\epsilon^2} \left( 1 + \frac{\|C\|_\infty}{\epsilon} \right)^2 \right).$$

We now provide analogous results for the RAGAS algorithm (cf. Algorithm 2).

**Lemma B.7** *Let* $\{(U_t, \pi_t)\}_{t \geq 1}$ *be the iterates generated by Algorithm 2. Then, we have*

$$\frac{1}{T} \left( \sum_{t=0}^{T-1} \|\text{grad } f_\eta(U_t)\|_F^2 \right) \leq \frac{8\|C\|_\infty \Delta_f}{\gamma T} + \frac{\epsilon^2}{10},$$

*where* $\Delta_f = \max_{U \in \text{St}(d,k)} f_\eta(U) - f_\eta(U_0)$ *is the initial objective gap.*

**Theorem B.8** *Letting* $\{(U_t, \pi_t)\}_{t \geq 1}$ *be the iterates generated by Algorithm 2, the number of iterations required to reach* $\text{dist}(0, \text{subdiff } f(U_t)) \leq \epsilon$ *satisfies*

$$t = \widetilde{O} \left( \frac{k\|C\|_\infty^2}{\epsilon^2} \left( 1 + \frac{\|C\|_\infty}{\epsilon} \right)^2 \right).$$

From Theorem B.6 and B.8, Algorithm 1 and 2 achieve the same iteration complexity. Furthermore, the number of arithmetic operations at each loop of Algorithm 1 and 2 are also the same. Thus, the complexity bound of Algorithm 2 is the same as that of Algorithm 1.

# C  Proofs

In this section, we present all of the remaining proofs.

## C.1  Proof of Lemma 2.1

By Vial [85, Proposition 4.3], it suffices to show that the function $f(U) - \|C\|_\infty \|U\|_F^2$ is concave for any $U \in \mathbb{R}^{d \times k}$. By the definition of $f$, we have

$$f(U) = \min_{\pi \in \Pi(\mu,\nu)} \text{Trace} \left( U^\top V_\pi U \right).$$

Since $\{x_1, x_2, \ldots, x_n\} \subseteq \mathbb{R}^d$ and $\{y_1, y_2, \ldots, y_n\} \subseteq \mathbb{R}^d$ are two given groups of $n$ atoms in $\mathbb{R}^d$, the coefficient matrix $C$ is independent of $U$ and $\pi$. Furthermore, $\sum_{i=1}^n \sum_{j=1}^n \pi_{i,j} = 1$ and $\pi_{i,j} \geq 0$ for all $i, j \in [n]$ since $\pi \in \Pi(\mu, \nu)$. Putting these pieces together with Jensen's inequality, we have

$$\|V_\pi\|_F \leq \sum_{i=1}^n \sum_{j=1}^n \pi_{i,j} \|(x_i - y_j)(x_i - y_j)^\top\|_F \leq \max_{1 \leq i,j \leq n} \|x_i - y_j\|^2 = \|C\|_\infty.$$

This implies that $U \mapsto \mathrm{Trace}(U^\top V_\pi U) - \|C\|_\infty \|U\|_F^2$ is concave for any $\pi \in \Pi(\mu, \nu)$. Since $\Pi(\mu, \nu)$ is compact, Danskin's theorem [72] implies the desired result.

## C.2 Proof of Lemma 2.2

By the definition of the subdifferential $\partial f$, it suffices to show that $\|V_\pi U\|_F \leq \|C\|_\infty$ for all $\pi \in \Pi(\mu, \nu)$ and $U \in \mathrm{St}(d, k)$. Indeed, by the definition, $V_\pi$ is symmetric and positive semi-definite. Therefore, we have

$$\max_{U \in \mathrm{St}(d,k)} \|V_\pi U\|_F \leq \|V_\pi\|_F \leq \|C\|_\infty.$$

Putting these pieces together yields the desired result.

## C.3 Proof of Lemma B.1

It is clear that we have $f_\eta(\bullet) = \min_{\pi \in \Pi(\mu, \nu)} g(\bullet, \pi)$. Furthermore, $\pi^\star(\bullet) = \mathrm{argmin}_{\pi \in \Pi(\mu, \nu)} g(\bullet, \pi)$ is uniquely defined. Putting these pieces with the compactness of $\Pi(\mu, \nu)$ and the smoothness of $g(\bullet, \pi)$, Danskin's theorem [72] implies $f_\eta$ is continuously differentiable and the gradient is

$$\nabla f_\eta(U) = 2V_{\pi^\star(U)} U \quad \text{for all } U \in \mathbb{R}^{d \times k}.$$

Since $U \in \mathrm{St}(d, k)$ and $\pi^\star(U) \in \Pi(\mu, \nu)$, we have

$$\|\nabla f_\eta(U)\|_F = 2\|V_{\pi^\star(U)} U\|_F \leq 2\|V_{\pi^\star(U)}\|_F \leq 2\|C\|_\infty.$$

This completes the proof.

## C.4 Proof of Lemma B.2

It suffices to prove that

$$\|\nabla f_\eta(\alpha U_1 + (1 - \alpha)U_2) - \nabla f_\eta(U_2)\|_F \leq \left(2\|C\|_\infty + \frac{4\|C\|_\infty^2}{\eta}\right)\alpha\|U_1 - U_2\|_F,$$

for any $U_1, U_2 \in \mathrm{St}(d, k)$ and any $\alpha \in [0, 1]$. Indeed, let $U_\alpha = \alpha U_1 + (1 - \alpha)U_2$, we have

$$\|\nabla f_\eta(U_\alpha) - \nabla f_\eta(U_2)\|_F \leq 2\|V_{\pi^\star(U_\alpha)}\|_F \|U_\alpha - U_2\|_F + 2\|V_{\pi^\star(U_\alpha)} - V_{\pi^\star(U_2)}\|_F.$$

Since $\pi^\star(U_\alpha) \in \Pi(\mu, \nu)$, we have $\|V_{\pi^\star(U_\alpha)}\|_F \leq \|C\|_\infty$. By the definition of $V_\pi$, we have

$$\|V_{\pi^\star(U_\alpha)} - V_{\pi^\star(U_2)}\|_F \leq \sum_{i=1}^n \sum_{j=1}^n |\pi_{i,j}^\star(U_\alpha) - \pi_{i,j}^\star(U_2)| \|x_i - y_j\|^2 \leq \|C\|_\infty \|\pi^\star(U_\alpha) - \pi^\star(U_2)\|_1.$$

Putting these pieces together yields that

$$\|\nabla f_\eta(U_\alpha) - \nabla f_\eta(U_2)\|_F \leq 2\|C\|_\infty \|U_\alpha - U_2\|_F + 2\|C\|_\infty \|\pi^\star(U_\alpha) - \pi^\star(U_2)\|_1. \quad \text{(C.1)}$$

Using the property of the entropy regularization $H(\bullet)$, we have $g(U, \bullet)$ is strongly convex with respect to $\ell_1$-norm and the module is $\eta$. This implies that

$$g(U_\alpha, \pi^\star(U_2)) - g(U_\alpha, \pi^\star(U_\alpha)) - \langle \nabla_\pi g(U_\alpha, \pi^\star(U_\alpha)), \pi^\star(U_2) - \pi^\star(U_\alpha) \rangle$$
$$\geq (\eta/2)\|\pi^\star(U_\alpha) - \pi^\star(U_2)\|_1^2,$$
$$g(U_\alpha, \pi^\star(U_\alpha)) - g(U_\alpha, \pi^\star(U_2)) - \langle \nabla_\pi g(U_\alpha, \pi^\star(U_2)), \pi^\star(U_\alpha) - \pi^\star(U_2) \rangle$$
$$\geq (\eta/2)\|\pi^\star(U_\alpha) - \pi^\star(U_2)\|_1^2.$$

Summing up these inequalities yields

$$\langle \nabla_\pi g(U_\alpha, \pi^\star(U_\alpha)) - \nabla_\pi g(U_\alpha, \pi^\star(U_2)), \pi^\star(U_\alpha) - \pi^\star(U_2) \rangle \geq \eta \|\pi^\star(U_\alpha) - \pi^\star(U_2)\|_1^2. \quad \text{(C.2)}$$

Furthermore, by the first-order optimality condition of $\pi^\star(U_1)$ and $\pi^\star(U_2)$, we have

$$\begin{aligned}
\langle \nabla_\pi g(U_\alpha, \pi^\star(U_\alpha)), \pi^\star(U_2) - \pi^\star(U_\alpha) \rangle &\geq 0, \\
\langle \nabla_\pi g(U_2, \pi^\star(U_2)), \pi^\star(U_\alpha) - \pi^\star(U_2) \rangle &\geq 0.
\end{aligned}$$

Summing up these inequalities yields

$$\langle \nabla_\pi g(U_2, \pi^\star(U_2)) - \nabla_\pi g(U_\alpha, \pi^\star(U_\alpha)), \pi^\star(U_\alpha) - \pi^\star(U_2) \rangle \geq 0. \quad \text{(C.3)}$$

Summing up Eq. (C.2) and Eq. (C.3) and further using Hölder's inequality, we have

$$\|\pi^\star(U_\alpha) - \pi^\star(U_2)\|_1 \leq (1/\eta)\|\nabla_\pi g(U_2, \pi^\star(U_2)) - \nabla_\pi g(U_\alpha, \pi^\star(U_2))\|_\infty.$$

By the definition of function $g$, we have

$$\begin{aligned}
\|\nabla_\pi g(U_2, \pi^\star(U_2)) - \nabla_\pi g(U_\alpha, \pi^\star(U_2))\|_\infty &\leq \max_{1 \leq i,j \leq n} |(x_i - x_j)^\top (U_2 U_2^\top - U_\alpha U_\alpha^\top)(x_i - x_j)| \\
&\leq \left( \max_{1 \leq i,j \leq n} \|x_i - y_j\|^2 \right) \|U_2 U_2^\top - U_\alpha U_\alpha^\top\|_F \\
&= \|C\|_\infty \|U_2 U_2^\top - U_\alpha U_\alpha^\top\|_F.
\end{aligned}$$

Since $U_1, U_2 \in \mathrm{St}(d, k)$, we have

$$\begin{aligned}
\|U_2 U_2^\top - U_\alpha U_\alpha^\top\|_F &\leq \|U_2(U_2 - U_\alpha)^\top\|_F + \|(U_2 - U_\alpha)U_\alpha^\top\|_F \\
&\leq \|U_2 - U_\alpha\|_F + \|(U_2 - U_\alpha)(\alpha U_1 + (1-\alpha)U_2)^\top\|_F \\
&\leq \|U_2 - U_\alpha\|_F + \alpha\|(U_2 - U_\alpha)U_1^\top\|_F + (1-\alpha)\|(U_2 - U_\alpha)U_2^\top\|_F \\
&\leq 2\|U_2 - U_\alpha\|_F.
\end{aligned}$$

Putting these pieces together yields that

$$\|\pi^\star(U_\alpha) - \pi^\star(U_2)\|_1 \leq \frac{2\|C\|_\infty}{\eta}\|U_\alpha - U_2\|_F. \quad \text{(C.4)}$$

Plugging Eq. (C.4) into Eq. (C.1) yields the desired result.

### C.5  Proof of Lemma B.5

Using Lemma B.2 with $U_1 = U_{t+1}$ and $U_2 = U_t$, we have

$$f_\eta(U_{t+1}) - f_\eta(U_t) - \langle \nabla f_\eta(U_t), U_{t+1} - U_t \rangle \geq -\left( \|C\|_\infty + \frac{2\|C\|_\infty^2}{\eta} \right) \|U_{t+1} - U_t\|_F^2. \quad \text{(C.5)}$$

By the definition of $U_{t+1}$, we have

$$\begin{aligned}
\langle \nabla f_\eta(U_t), U_{t+1} - U_t \rangle &= \langle \nabla f_\eta(U_t), \mathrm{Retr}_{U_t}(\gamma \xi_{t+1}) - U_t \rangle \\
&= \langle \nabla f_\eta(U_t), \gamma \xi_{t+1} \rangle + \langle \nabla f_\eta(U_t), \mathrm{Retr}_{U_t}(\gamma \xi_{t+1}) - (U_t + \gamma \xi_{t+1}) \rangle \\
&\geq \langle \nabla f_\eta(U_t), \gamma \xi_{t+1} \rangle - \|\nabla f_\eta(U_t)\|_F \|\mathrm{Retr}_{U_t}(\gamma \xi_{t+1}) - (U_t + \gamma \xi_{t+1})\|_F.
\end{aligned}$$

By Lemma B.1, we have $\|\nabla f_\eta(U)\|_F \leq 2\|C\|_\infty$. Putting these pieces with Proposition A.1 yields that

$$\langle \nabla f_\eta(U_t), U_{t+1} - U_t \rangle \geq \gamma \langle \nabla f_\eta(U_t), \xi_{t+1} \rangle - 2\gamma^2 L_2 \|C\|_\infty \|\xi_{t+1}\|_F^2. \quad \text{(C.6)}$$

Using Proposition A.1 again, we have

$$\|U_{t+1} - U_t\|_F^2 = \|\mathrm{Retr}_{U_t}(\gamma \xi_{t+1}) - U_t\|_F^2 \leq \gamma^2 L_1^2 \|\xi_{t+1}\|_F^2. \quad \text{(C.7)}$$

Combining Eq. (C.5), Eq. (C.6) and Eq. (C.7) yields

$$f_\eta(U_{t+1}) - f_\eta(U_t) \geq \gamma \langle \nabla f_\eta(U_t), \xi_{t+1} \rangle - \gamma^2((L_1^2 + 2L_2)\|C\|_\infty + 2\eta^{-1}L_1^2\|C\|_\infty^2)\|\xi_{t+1}\|_F^2. \quad \text{(C.8)}$$

Recall that $\mathrm{grad}\, f_\eta(U_t) = P_{\mathrm{T}_{U_t}\mathrm{St}}(\nabla f_\eta(U_t))$ and $\xi_{t+1} = P_{\mathrm{T}_{U_t}\mathrm{St}}(2V_{\pi_{t+1}}U_t)$, we have

$$\langle \nabla f_\eta(U_t), \xi_{t+1} \rangle = \langle \mathrm{grad}\, f_\eta(U_t), \xi_{t+1} \rangle = \|\mathrm{grad}\, f_\eta(U_t)\|_F^2 + \langle \mathrm{grad}\, f_\eta(U_t), \xi_{t+1} - \mathrm{grad}\, f_\eta(U_t) \rangle$$

Using Young's inequality, we have

$$\langle \nabla f_\eta(U_t), \xi_{t+1} \rangle \geq (1/2) \left( \|\mathrm{grad}\, f_\eta(U_t)\|_F^2 - \|\xi_{t+1} - \mathrm{grad}\, f_\eta(U_t)\|_F^2 \right).$$

Furthermore, we have $\|\xi_{t+1}\|_F^2 \leq 2\|\mathrm{grad}\, f_\eta(U_t)\|_F^2 + 2\|\xi_{t+1} - \mathrm{grad}\, f_\eta(U_t)\|_F^2$. Putting these pieces together with Eq. (C.8) yields that

$$f_\eta(U_{t+1}) - f_\eta(U_t) \geq \gamma \left( \frac{1}{2} - \gamma(2L_1^2\|C\|_\infty + 4L_2\|C\|_\infty + 4\eta^{-1}L_1^2\|C\|_\infty^2) \right) \|\mathrm{grad}\, f_\eta(U_t)\|_F^2$$

$$-\gamma \left( \frac{1}{2} + \gamma(2L_1^2\|C\|_\infty + 4L_2\|C\|_\infty + 4\eta^{-1}L_1^2\|C\|_\infty^2) \right) \|\xi_{t+1} - \mathrm{grad}\, f_\eta(U_t)\|_F^2. \quad \text{(C.9)}$$

Since $\xi_{t+1} = P_{\mathrm{T}_{U_t}\mathrm{St}}(2V_{\pi_{t+1}}U_t)$ and $\mathrm{grad}\, f_\eta(U_t) = P_{\mathrm{T}_{U_t}\mathrm{St}}(2V_{\tilde{\pi}_t^\star}U_t)$ where $\tilde{\pi}_t^\star$ is a minimizer of the entropic regularized OT problem, i.e., $\tilde{\pi}_t^\star \in \mathrm{argmin}_{\pi \in \Pi(\mu,\nu)} \{\langle U_t U_t^\top, V_\pi \rangle - \eta H(\pi)\}$, we have

$$\|\xi_{t+1} - \mathrm{grad}\, f_\eta(U_t)\|_F \leq 2\|(V_{\pi_{t+1}} - V_{\tilde{\pi}_t^\star})U_t\|_F = 2\|V_{\pi_{t+1}} - V_{\tilde{\pi}_t^\star}\|_F.$$

By the definition of $V_\pi$ and using the stopping criterion: $\|\pi_{t+1} - \tilde{\pi}_t^\star\|_1 \leq \hat{\epsilon} = \frac{\epsilon}{10\|C\|_\infty}$, we have

$$\|V_{\pi_{t+1}} - V_{\tilde{\pi}_t^\star}\|_F \leq \|C\|_\infty \|\pi_{t+1} - \tilde{\pi}_t^\star\|_1 \leq \frac{\epsilon}{10}.$$

Putting these pieces together yields that

$$\|\xi_{t+1} - \mathrm{grad}\, f_\eta(U_t)\|_F \leq \frac{\epsilon}{5}. \quad \text{(C.10)}$$

Plugging Eq. (C.10) into Eq. (C.9) with the definition of $\gamma$ yields that

$$f_\eta(U_{t+1}) - f_\eta(U_t) \geq \frac{\gamma\|\mathrm{grad}\, f_\eta(U_t)\|_F^2}{4} - \frac{\gamma\epsilon^2}{20}.$$

Summing and rearranging the resulting inequality yields that

$$\frac{1}{T} \left( \sum_{t=0}^{T-1} \|\mathrm{grad}\, f_\eta(U_t)\|_F^2 \right) \leq \frac{4(f_\eta(U_T) - f_\eta(U_0))}{\gamma T} + \frac{\epsilon^2}{5}.$$

This together with the definition of $\Delta_f$ implies the desired result.

## C.6   Proof of Theorem B.6

Let $\tilde{\pi}_t^\star$ be a minimzer of entropy-regularized OT problem and $\pi_t^\star$ be the projection of $\tilde{\pi}_t^\star$ onto the optimal solution set of unregularized OT problem. More specifically, the unregularized OT problem is a LP and the optimal solution set is a polyhedron set ($t^\star$ is an optimal objective value)

$$\mathcal{S} = \{\pi \in \mathbb{R}^{d \times d} \mid \pi \in \Pi(\mu,\nu), \langle U_t U_t^\top, V_\pi \rangle = t^\star\}.$$

Then we have

$$\tilde{\pi}_t^\star \in \underset{\pi \in \Pi(\mu,\nu)}{\mathrm{argmin}}\ \langle U_t U_t^\top, V_\pi \rangle - \eta H(\pi), \qquad \pi_t^\star = \mathrm{proj}(\tilde{\pi}_t^\star) \in \underset{\pi \in \Pi(\mu,\nu)}{\mathrm{argmin}}\ \langle U_t U_t^\top, V_\pi \rangle.$$

By definition, we have $\nabla f_\eta(U_t) = 2V_{\tilde{\pi}_t^\star}U_t$ and $2V_{\pi_t^\star}U_t \in \partial f(U_t)$. This together with the definition of Riemannian gradient and Riemannian subdifferential yields that

$$\begin{aligned} \mathrm{grad}\, f_\eta(U_t) &= P_{\mathrm{T}_{U_t}\mathrm{St}}(2V_{\tilde{\pi}_t^\star}U_t), \\ \mathrm{subdiff}\, f(U_t) &\ni P_{\mathrm{T}_{U_t}\mathrm{St}}(2V_{\pi_t^\star}U_t). \end{aligned}$$

Therefore, we conclude that

$$\begin{aligned} \mathrm{dist}(0, \mathrm{subdiff}\, f(U_t)) &\leq \|P_{\mathrm{T}_{U_t}\mathrm{St}}(2V_{\pi_t^\star}U_t)\|_F \\ &\leq \|P_{\mathrm{T}_{U_t}\mathrm{St}}(2V_{\tilde{\pi}_t^\star}U_t)\|_F + \|P_{\mathrm{T}_{U_t}\mathrm{St}}(2V_{\pi_t^\star}U_t) - P_{\mathrm{T}_{U_t}\mathrm{St}}(2V_{\tilde{\pi}_t^\star}U_t)\|_F \\ &\leq \|\mathrm{grad}\, f_\eta(U_t)\|_F + 2\|(V_{\pi_t^\star} - V_{\tilde{\pi}_t^\star})U_t\|_F. \end{aligned}$$

Note that scaling the objective function by $\|C\|_\infty$ will not change the optimal solution set. Since $U_t \in \mathrm{St}(d,k)$, each entry of the coefficient in the normalized objective function is less than 1. By Lemma B.4, we obtain that there exists a constant $\bar{\theta}$ independent of $\|C\|_\infty$ such that

$$\|\tilde{\pi}_t^\star - \pi_t^\star\|_1 \;\leq\; \bar{\theta}\left\|\left\langle U_t U_t^\top, \frac{V_{\tilde{\pi}_t^\star} - V_{\pi_t^\star}}{\|C\|_\infty}\right\rangle\right\|_1.$$

By the definition of $\tilde{\pi}_t^\star$, we have $\langle U_t U_t^\top, V_{\tilde{\pi}_t^\star}\rangle - \eta H(\tilde{\pi}_t^\star) \leq \langle U_t U_t^\top, V_{\pi_t^\star}\rangle - \eta H(\pi_t^\star)$. Since $0 \leq H(\pi) \leq 2\log(n)$ and $\eta = \frac{\epsilon \min\{1, 1/\bar{\theta}\}}{40\log(n)}$, we have

$$\tilde{\pi}_t^\star \;\in\; \Pi(\mu,\nu), \qquad 0 \;\leq\; \langle U_t U_t^\top, V_{\tilde{\pi}_t^\star} - V_{\pi_t^\star}\rangle \;\leq\; \epsilon/(20\bar{\theta}).$$

Putting these pieces together yields that

$$\|\tilde{\pi}_t^\star - \pi_t^\star\|_1 \;\leq\; \frac{\epsilon}{20\|C\|_\infty \bar{\theta}}.$$

By the definition of $U_t$ and $V_\pi$, we have

$$\|(V_{\pi_t^\star} - V_{\tilde{\pi}_t^\star})U_t\|_F \;=\; \|V_{\pi_t^\star} - V_{\tilde{\pi}_t^\star}\|_F \;\leq\; \bar{\theta}\|C\|_\infty \|\tilde{\pi}_t^\star - \pi_t^\star\|_1 \;\leq\; \frac{\epsilon}{20}.$$

Putting these pieces together yields

$$\mathrm{dist}(0, \mathrm{subdiff}\, f(U_t)) \;\leq\; \|\mathrm{grad}\, f_\eta(U_t)\|_F + \frac{\epsilon}{10}.$$

Combining this inequality with Lemma B.5 and the Cauchy-Schwarz inequality, we have

$$\frac{1}{T}\left(\sum_{t=0}^{T-1}[\mathrm{dist}(0, \mathrm{subdiff}\, f(U_t))]^2\right) \;\leq\; \frac{2}{T}\left(\sum_{t=0}^{T-1}\|\mathrm{grad}\, f_\eta(U_t)\|_F^2\right) + \frac{\epsilon^2}{50} \;\leq\; \frac{8\Delta_f}{\gamma T} + \frac{2\epsilon^2}{5} + \frac{\epsilon^2}{50}$$

$$\leq\; \frac{8\Delta_f}{\gamma T} + \frac{\epsilon^2}{2}.$$

Given that $\mathrm{dist}(0, \mathrm{subdiff}\, f(U_t)) > \epsilon$ for all $t = 0, 1, \ldots, T-1$ and

$$\frac{1}{\gamma} \;=\; (8L_1^2 + 16L_2)\|C\|_\infty + \frac{16L_1^2\|C\|_\infty^2}{\eta} \;=\; (8L_1^2 + 16L_2)\|C\|_\infty + \frac{640L_1^2 \max\{1,\bar{\theta}\}\|C\|_\infty^2 \log(n)}{\epsilon}.$$

we conclude that the upper bound $T$ must satisfy

$$\epsilon^2 \;\leq\; \frac{16\Delta_f}{T}\left((8L_1^2 + 16L_2)\|C\|_\infty + \frac{640L_1^2 \max\{1,\bar{\theta}\}\|C\|_\infty^2 \log(n)}{\epsilon}\right).$$

Using Lemma B.2, we have

$$\Delta_f \;\leq\; \left(\|C\|_\infty + \frac{2\|C\|_\infty^2}{\eta}\right)\left(\max_{U\in\mathrm{St}(d,k)}\|U - U_0\|_F^2\right) \;=\; k\left(2\|C\|_\infty + \frac{4\|C\|_\infty^2}{\eta}\right)$$

$$=\; k\left(2\|C\|_\infty + \frac{160\max\{1,\bar{\theta}\}\|C\|_\infty^2 \log(n)}{\epsilon}\right).$$

Putting these pieces together implies the desired result.

### C.7   Proof of Lemma B.7

Using the same argument as in the proof of Lemma B.5, we have

$$f_\eta(U_{t+1}) - f_\eta(U_t) \;\geq\; \gamma\langle \nabla f_\eta(U_t), \xi_{t+1}\rangle - \gamma^2((L_1^2 + 2L_2)\|C\|_\infty + 2\eta^{-1}L_1^2\|C\|_\infty^2)\|\xi_{t+1}\|_F^2. \tag{C.11}$$

Recall that $\mathrm{grad}\, f_\eta(U_t) = P_{\mathrm{T}_{U_t}\mathrm{St}}(\nabla f_\eta(U_t))$ and the definition of $\xi_{t+1}$, we have

$$\langle \nabla f_\eta(U_t), \xi_{t+1}\rangle \;=\; \langle \mathrm{grad}\, f_\eta(U_t), \xi_{t+1}\rangle$$

$$=\; \langle \mathrm{grad}\, f_\eta(U_t), \mathrm{Diag}\,(\widehat{p}_{t+1})^{-1/4}(\mathrm{grad}\, f_\eta(U_t))\mathrm{Diag}\,(\widehat{q}_{t+1})^{-1/4}\rangle$$

$$+\langle \mathrm{grad}\, f_\eta(U_t), \mathrm{Diag}\,(\widehat{p}_{t+1})^{-1/4}(G_{t+1} - \mathrm{grad}\, f_\eta(U_t))\mathrm{Diag}\,(\widehat{q}_{t+1})^{-1/4}\rangle.$$

Using the Cauchy-Schwarz inequality and the nonexpansiveness of $P_{\mathrm{T}_{U_t}\mathrm{St}}$, we have

$$
\begin{aligned}
\|\xi_{t+1}\|_F^2 \quad \leq \quad & 2\|P_{\mathrm{T}_{U_t}\mathrm{St}}(\mathrm{Diag}\,(\widehat{p}_{t+1})^{-1/4}(\mathrm{grad}\,f_\eta(U_t))\mathrm{Diag}\,(\widehat{q}_{t+1})^{-1/4})\|_F^2 \\
& +2\|\xi_{t+1} - P_{\mathrm{T}_{U_t}\mathrm{St}}(\mathrm{Diag}\,(\widehat{p}_{t+1})^{-1/4}(\mathrm{grad}\,f_\eta(U_t))\mathrm{Diag}\,(\widehat{q}_{t+1})^{-1/4})\|_F^2 \\
\leq \quad & 2\|\mathrm{Diag}\,(\widehat{p}_{t+1})^{-1/4}(\mathrm{grad}\,f_\eta(U_t))\mathrm{Diag}\,(\widehat{q}_{t+1})^{-1/4}\|_F^2 \\
& +2\|\mathrm{Diag}\,(\widehat{p}_{t+1})^{-1/4}(G_{t+1} - \mathrm{grad}\,f_\eta(U_t))\mathrm{Diag}\,(\widehat{q}_{t+1})^{-1/4}\|_F^2.
\end{aligned}
$$

Furthermore, by the definition of $G_{t+1}$, we have $\|G_{t+1}\|_F \leq 2\|C\|_\infty$ and hence

$$
\mathbf{0}_d \leq \frac{\mathrm{diag}(G_{t+1}G_{t+1}^\top)}{k} \leq 4\|C\|_\infty^2 \mathbf{1}_d, \qquad \mathbf{0}_k \leq \frac{\mathrm{diag}(G_{t+1}^\top G_{t+1})}{d} \preceq 4\|C\|_\infty^2 \mathbf{1}_k.
$$

By the definition of $p_t$ and $q_t$, we have $\mathbf{0}_d \preceq p_t \preceq 4\|C\|_\infty^2 \mathbf{1}_d$ and $\mathbf{0}_k \preceq q_t \preceq 4\|C\|_\infty^2 \mathbf{1}_k$. This together with the definition of $\widehat{p}_t$ and $\widehat{q}_t$ implies that

$$
\alpha\|C\|_\infty^2 \mathbf{1}_d \leq \widehat{p}_t \leq 4\|C\|_\infty^2 \mathbf{1}_d, \qquad \alpha\|C\|_\infty^2 \mathbf{1}_k \leq \widehat{q}_t \leq 4\|C\|_\infty^2 \mathbf{1}_k.
$$

This inequality together with Young's inequality implies that

$$
\begin{aligned}
\langle \nabla f_\eta(U_t), \xi_{t+1}\rangle \quad \geq \quad & \frac{\|\mathrm{grad}\,f_\eta(U_t)\|_F^2}{2\|C\|_\infty} - \frac{1}{\sqrt{\alpha}\|C\|_\infty}\left(\frac{\sqrt{\alpha}\|\mathrm{grad}\,f_\eta(U_t)\|_F^2}{4} + \frac{\|G_{t+1} - \mathrm{grad}\,f_\eta(U_t)\|_F^2}{\sqrt{\alpha}}\right) \\
= \quad & \frac{\|\mathrm{grad}\,f_\eta(U_t)\|_F^2}{4\|C\|_\infty} - \frac{\|G_{t+1} - \mathrm{grad}\,f_\eta(U_t)\|_F^2}{\alpha\|C\|_\infty},
\end{aligned}
$$

and

$$
\|\xi_{t+1}\|_F^2 \quad \leq \quad \frac{2\|\mathrm{grad}\,f_\eta(U_t)\|_F^2}{\alpha\|C\|_\infty^2} + \frac{2\|G_{t+1} - \mathrm{grad}\,f_\eta(U_t)\|_F^2}{\alpha\|C\|_\infty^2}.
$$

Putting these pieces together with Eq. (C.11) yields that

$$
\begin{aligned}
f_\eta(U_{t+1}) - f_\eta(U_t) \quad \geq \quad & \frac{\gamma}{4\|C\|_\infty}\left(1 - \frac{8\gamma}{\alpha}\left(L_1^2 + 2L_2 + 2\eta^{-1}L_1^2\|C\|_\infty\right)\right)\|\mathrm{grad}\,f_\eta(U_t)\|_F^2 \\
& -\frac{\gamma}{\alpha\|C\|_\infty}\left(1 + \gamma(2L_1^2 + 4L_2 + 4\eta^{-1}L_1^2\|C\|_\infty)\right)\|G_{t+1} - \mathrm{grad}\,f_\eta(U_t)\|_F^2. \quad \text{(C.12)}
\end{aligned}
$$

Recall that $G_{t+1} = P_{\mathrm{T}_{U_t}\mathrm{St}}(2V_{\pi_{t+1}}U_t)$ and $\mathrm{grad}\,f_\eta(U_t) = P_{\mathrm{T}_{U_t}\mathrm{St}}(2V_{\widetilde{\pi}_t^\star}U_t)$. Then we can apply the same argument as in the proof of Lemma B.5 and obtain that

$$
\|G_{t+1} - \mathrm{grad}\,f_\eta(U_t)\|_F \quad \leq \quad \frac{\epsilon\sqrt{\alpha}}{10}. \quad \text{(C.13)}
$$

Plugging Eq. (C.13) into Eq. (C.12) with the definition of $\gamma$ yields that

$$
f_\eta(U_{t+1}) - f_\eta(U_t) \quad \geq \quad \frac{\gamma\|\mathrm{grad}\,f_\eta(U_t)\|_F^2}{8\|C\|_\infty} - \frac{\gamma\epsilon^2}{80\|C\|_\infty}.
$$

Summing and rearranging the resulting inequality yields that

$$
\frac{1}{T}\left(\sum_{t=0}^{T-1}\|\mathrm{grad}\,f_\eta(U_t)\|_F^2\right) \quad \leq \quad \frac{8\|C\|_\infty(f_\eta(U_T) - f_\eta(U_0))}{\gamma T} + \frac{\epsilon^2}{10}.
$$

This together with the definition of $\Delta_f$ implies the desired result.

## C.8 Proof of Theorem B.8

Using the same argument as in the proof of Theorem B.6, we have

$$
\mathrm{dist}(0, \mathrm{subdiff}\,f(U_t)) \quad \leq \quad \|\mathrm{grad}\,f_\eta(U_t)\|_F + \frac{\epsilon}{10}.
$$

Combining this inequality with Lemma B.7 and the Cauchy-Schwarz inequality, we have

$$
\begin{aligned}
\frac{1}{T}\left(\sum_{t=0}^{T-1}[\mathrm{dist}(0, \mathrm{subdiff}\,f(U_t))]^2\right) \quad \leq \quad & \frac{2}{T}\left(\sum_{t=0}^{T-1}\|\mathrm{grad}\,f_\eta(U_t)\|_F^2\right) + \frac{\epsilon^2}{50} \\
\leq \quad & \frac{16\|C\|_\infty\Delta_f}{\gamma T} + \frac{\epsilon^2}{5} + \frac{\epsilon^2}{50} \leq \frac{16\|C\|_\infty\Delta_f}{\gamma T} + \frac{\epsilon^2}{2}.
\end{aligned}
$$

Given that $\text{dist}(0, \text{subdiff } f(U_t)) > \epsilon$ for all $t = 0, 1, \ldots, T-1$ and

$$\frac{1}{\gamma} = 16L_1^2 + 32L_2 + \frac{1280L_1^2 \max\{1, \bar{\theta}\}\|C\|_\infty \log(n)}{\epsilon},$$

we conclude that the upper bound $T$ must satisfies

$$\epsilon^2 \leq \frac{64\|C\|_\infty \Delta_f}{T} \left(16L_1^2 + 32L_2 + \frac{1280L_1^2 \max\{1, \bar{\theta}\}\|C\|_\infty \log(n)}{\epsilon}\right).$$

Using Lemma B.2, we have

$$\begin{aligned}
\Delta_f &\leq \left(\|C\|_\infty + \frac{2\|C\|_\infty^2}{\eta}\right)\left(\max_{U \in \text{St}(d,k)} \|U - U_0\|_F^2\right) = k\left(2\|C\|_\infty + \frac{4\|C\|_\infty^2}{\eta}\right) \\
&= k\left(2\|C\|_\infty + \frac{160 \max\{1, \bar{\theta}\}\|C\|_\infty^2 \log(n)}{\epsilon}\right).
\end{aligned}$$

Putting these pieces together implies the desired result.

## C.9 Proof of Theorem 3.1

First, Theorem B.6 and B.8 imply that both algorithms achieve the same the iteration complexity as follows,

$$t = \widetilde{O}\left(\frac{k\|C\|_\infty^2}{\epsilon^2}\left(1 + \frac{\|C\|_\infty}{\epsilon}\right)^2\right). \tag{C.14}$$

This implies that $U_t$ is an $\epsilon$-approximate optimal subspace projection of problem (2.7). By the definition of $\widehat{\epsilon}$ and using the stopping criterion of the subroutine $\textsc{REGOT}(\{(x_i, r_i)\}_{i\in[n]}, \{(y_j, c_j)\}_{j\in[n]}, U_t, \eta, \widehat{\epsilon})$, we have $\pi_{t+1} \in \Pi(\mu, \nu)$ and

$$0 \leq \langle U_t U_t^\top, V_{\pi_{t+1}} - V_{\tilde{\pi}_t^\star}\rangle \leq \|C\|_\infty \|\pi_{t+1} - \tilde{\pi}_t^\star\|_1 \leq \|C\|_\infty \widehat{\epsilon} \leq \epsilon/2.$$

where $\tilde{\pi}_t^\star$ is an unique minimzer of entropy-regularized OT problem. Furthermore, by the definition of $\tilde{\pi}_t^\star$, we have $\langle U_t U_t^\top, V_{\tilde{\pi}_t^\star}\rangle - \eta H(\tilde{\pi}_t^\star) \leq \langle U_t U_t^\top, V_{\pi_t^\star}\rangle - \eta H(\pi_t^\star)$. Since $0 \leq H(\pi) \leq 2\log(n)$ and $\eta = \frac{\epsilon \min\{1, 1/\bar{\theta}\}}{40 \log(n)}$, we have

$$\tilde{\pi}_t^\star \in \Pi(\mu, \nu), \qquad 0 \leq \langle U_t U_t^\top, V_{\tilde{\pi}_t^\star} - V_{\pi_t^\star}\rangle \leq \epsilon/2.$$

Putting these pieces together yields that $\pi_{t+1}$ is an $\epsilon$-approximate optimal transportation plan for the subspace projection $U_t$. Therefore, we conclude that $(U_t, \pi_{t+1}) \in \text{St}(d,k) \times \Pi(\mu, \nu)$ is an *$\epsilon$-approximate pair of optimal subspace projection and optimal transportation plan* of problem (2.3).

The remaining step is to analyze the complexity bound. Indeed, we first claim that the number of arithmetic operations required by the Sinkhorn iteration at each loop is upper bounded by

$$\widetilde{O}\left(\frac{n^2\|C\|_\infty^4}{\epsilon^4} + \frac{n^2\|C\|_\infty^8}{\epsilon^8}\right). \tag{C.15}$$

Furthermore, while **Step 5** and **Step 6** in Algorithm 1 can be implemented in $O(dk^2 + k^3)$ arithmetic operations, we still need to construct $V_{\pi_{t+1}}U_t$. A naive approach suggests to first construct $V_{\pi_{t+1}}$ using $O(n^2 dk)$ arithmetic operations and then perform the matrix multiplication using $O(d^2 k)$ arithmetic operations. This is computationally prohibitive since $d$ can be very large in practice. In contrast, we observe that

$$V_{\pi_{t+1}}U_t = \sum_{i=1}^n \sum_{j=1}^n (\pi_{t+1})_{i,j}(x_i - y_j)(x_i - y_j)^\top U_t.$$

Since $x_i - y_j \in \mathbb{R}^d$, it will take $O(dk)$ arithmetic operations for computing $(x_i - y_j)(x_i - y_j)^\top U_t$ for all $(i, j) \in [n] \times n$. This implies that the total number of arithmetic operations is $O(n^2 dk)$. Therefore, the number of arithmetic operations at each loop is

$$\widetilde{O}\left(n^2 dk + dk^2 + k^3 + \frac{n^2\|C\|_\infty^4}{\epsilon^4} + \frac{n^2\|C\|_\infty^8}{\epsilon^8}\right). \tag{C.16}$$

Putting Eq. (C.14) and Eq. (C.16) together with $k = \widetilde{O}(1)$ yields the desired result.

**Proof of claim** (C.15). The proof is based on the combination of several existing results proved by Altschuler et al. [3] and Dvurechensky et al. [30]. For the sake of completeness, we provide the details. More specifically, we consider solving the entropic regularized OT problem as follows,

$$\min_{\pi \in \mathbb{R}_+^{n \times n}} \langle C, \pi \rangle - \eta H(\pi), \quad \text{s.t. } r(\pi) = r, \ c(\pi) = c.$$

We leverage the Sinkhorn iteration which aims at maximizing the following function

$$f(u,v) = \mathbf{1}_n^\top B(u,v)\mathbf{1}_n - \langle u, r \rangle - \langle v, c \rangle, \quad \text{where } B(u,v) := \text{diag}(u)e^{-\frac{C}{\eta}}\text{diag}(v).$$

From the update scheme of Sinkhorn iteration, it is clear that $\mathbf{1}_n^\top B(u_j, v_j)\mathbf{1}_n = 1$ for each iteration $j$. By a straightforward calculation, we have

$$\langle C, B(u_j, v_j) \rangle - \eta H(B(u_j, v_j)) - (\langle C, B(u^\star, v^\star) \rangle - \eta H(B(u^\star, v^\star)))$$
$$\leq \eta(f(u^\star, v^\star) - f(u_j, v_j)) + \eta R(\|r(B(u_j, v_j)) - r\|_1 + \|c(B(u_j, v_j)) - c\|_1)$$

where $(u^\star, v^\star)$ is a maximizer of $f(u,v)$ over $\mathbb{R}^n \times \mathbb{R}^n$ and $R > 0$ is defined in Dvurechensky et al. [30, Lemma 1]. Since the entropic regularization function is strongly convex with respect to $\ell_1$-norm over the probability simplex and $B(u_j, v_j)$ can be vectorized as a probability vector, we have

$$\|B(u_j, v_j) - B(u^\star, v^\star)\|_1^2 \leq 2(f(u^\star, v^\star) - f(u_j, v_j)) + 2R(\|r(B(u_j, v_j)) - r\|_1 + \|c(B(u_j, v_j)) - c\|_1).$$

On one hand, by the definition of $(u^\star, v^\star)$ and $B(\cdot, \cdot)$, it is clear that $B(u^\star, v^\star)$ is an unique optimal solution of the entropic regularized OT problem and we further denote it as $\tilde{\pi}^\star$. On the other hand, the final output $\pi \in \Pi(\mu, \nu)$ is achieved by rounding $B(u_j, v_j)$ to $\Pi(\mu, \nu)$ for some $j$ using Altschuler et al. [3, Algorithm 2] and Altschuler et al. [3, Lemma 7] guarantees that

$$\|\tilde{\pi} - B(u_j, v_j)\|_1 \leq 2(\|r(B(u_j, v_j)) - r\|_1 + \|c(B(u_j, v_j)) - c\|_1).$$

Again, from the update scheme of Sinkhorn iteration and By Pinsker's inequality, we have

$$\sqrt{2(f(u^\star, v^\star) - f(u_j, v_j))} \geq \|r(B(u_j, v_j)) - r\|_1 + \|c(B(u_j, v_j)) - c\|_1.$$

Putting these pieces together yields that

$$\|\tilde{\pi} - \tilde{\pi}^\star\|_1 \leq c_1 (f(u^\star, v^\star) - f(u_j, v_j))^{1/2} + c_2\sqrt{R} (f(u^\star, v^\star) - f(u_j, v_j))^{1/4}$$

where $c_1, c_2 > 0$ are constants. Then, by using Eq.(12) in Dvurechensky et al. [30, Theorem 1], we have $f(u^\star, v^\star) - f(u_j, v_j) \leq \frac{2R^2}{j}$. This together with the definition of $R$ yields that the number of iterations required by the Sinkhorn iteration is

$$\widetilde{O}\left(\frac{\|C\|_\infty^4}{\epsilon^4} + \frac{\|C\|_\infty^8}{\epsilon^8}\right).$$

This completes the proof.

# D   Further Background Materials on Riemannian Optimization

The problem of optimizing a smooth function over the Riemannian manifold has been the subject of a large literature. Absil et al. [2] provide a comprehensive treatment, showing how first-order and second-order algorithms are extended to the Riemannian setting and proving asymptotic convergence to first-order stationary points. Boumal et al. [16] have established global sublinear convergence results for Riemannian gradient descent and Riemannian trust region algorithms, and further showed that the latter approach converges to a second order stationary point in polynomial time; see also Kasai and Mishra [49], Hu et al. [46, 47]. In contradistinction to the Euclidean setting, the Riemannian trust region algorithm requires a Hessian oracle. There have been also several recent papers on problem-specific algorithms [90, 37, 60] and primal-dual algorithms [95] for Riemannian optimization.

Compared to the smooth setting, Riemannian nonsmooth optimization is harder and relatively less explored [1]. There are two main lines of work. In the first category, one considers optimizing geodesically convex function over a Riemannian manifold with subgradient-type algorithms; see, e.g., Ferreira and Oliveira [32], Zhang and Sra [93], Bento et al. [7]. In particular, Ferreira and Oliveira [32] first established an asymptotic convergence result while Zhang and Sra [93], Bento

et al. [7] derived a global convergence rate of $O(\epsilon^{-2})$ for the Riemannian subgradient algorithm. Unfortunately, these results are not useful for understanding the computation of the PRW distance in Eq. (2.3) since the Stiefel manifold is *compact* and every continuous and geodesically convex function on a compact Riemannian manifold must be a constant; see Bishop and O'Neill [10, Proposition 2.2]. In the second category, one assumes the tractable computation of the proximal mapping of the objective function over the Riemannian manifold. Ferreira and Oliveira [33] proved that the Riemannian proximal point algorithm converges globally at a sublinear rate.

When specialized to the Stiefel manifold, Chen et al. [18] consider the composite objective and proposed to compute the proximal mapping of nonsmooth component function over the tangent space. The resulting Riemannian proximal gradient algorithm is practical in real applications while achieving theoretical guarantees. Li et al. [55] extended the results in Davis and Drusvyatskiy [27] to the Riemannian setting and proposed a family of Riemannian subgradienttype methods for optimizing a weakly convex function over the Stiefel manifold. They also proved that their algorithms have an iteration complexity of $O(\epsilon^{-4})$ for driving a near-optimal stationarity measure below $\epsilon$. Following up the direction proposed by Li et al. [55], we derive a near-optimal condition (Definition E.1 and E.2) for the max-min optimization model in Eq. (2.4) and propose an algorithm with the finite-time convergence under this stationarity measure.

Finally, there are several results on stochastic optimization over the Riemannian manifold. Bonnabel [12] proved the first asymptotic convergence result for Riemannian stochastic gradient descent, which is further extended by Zhang et al. [94], Tripuraneni et al. [84], Becigneul and Ganea [5]. If the Riemannian Hessian is not positive definite, a few recent works have developed frameworks to escape saddle points [81, 22].

# E   Near-Optimality Condition

In this section, we derive a near-optimal condition (Definition E.1 and E.2) for the max-min optimization model in Eq. (2.3) and the maximization of $f$ over $\mathrm{St}(d,k)$ in Eq. (2.7). Following Davis and Drusvyatskiy [27], Li et al. [55], we define the proximal mapping of $f$ over $\mathrm{St}(d,k)$ in Eq. (2.7), which takes into account both the Stiefel manifold constraint and max-min structure[5]:

$$p(U) \in \underset{\bar{U} \in \mathrm{St}(d,k)}{\operatorname{argmax}} \left\{ f(\bar{U}) - 6\|C\|_\infty \|\bar{U} - U\|_F^2 \right\} \quad \text{for all } U \in \mathrm{St}(d,k).$$

After a simple calculation, we have

$$\Theta(U) := 12\|C\|_\infty \|p(U) - U\|_F \geq \operatorname{dist}(0, \operatorname{subdiff} f(\operatorname{prox}_{\rho f}(U))),$$

Therefore, we conclude from Definition 2.6 that $p(U) \in \mathrm{St}(d,k)$ is $\epsilon$-approximate optimal subspace projection of $f$ over $\mathrm{St}(d,k)$ in Eq. (2.7) if $\Theta(U) \leq \epsilon$. We remark that $\Theta(\bullet)$ is a well-defined surrogate stationarity measure of $f$ over $\mathrm{St}(d,k)$ in Eq. (2.7). Indeed, if $\Theta(U) = 0$, then $U \in \mathrm{St}(d,k)$ is an optimal subspace projection. This inspires the following $\epsilon$-near-optimality condition for any $\widehat{U} \in \mathrm{St}(d,k)$.

**Definition E.1** *A subspace projection $\widehat{U} \in \mathrm{St}(d,k)$ is called an $\epsilon$-approximate near-optimal subspace projection of $f$ over $\mathrm{St}(d,k)$ in Eq. (2.7) if it satisfies $\Theta(\widehat{U}) \leq \epsilon$.*

Equipped with Definition 2.2 and E.1, we define an $\epsilon$-approximate pair of near-optimal subspace projection and optimal transportation plan for the computation of the PRW distance in Eq. (2.3).

**Definition E.2** *The pair of subspace projection and transportation plan $(\widehat{U}, \widehat{\pi}) \in \mathrm{St}(d,k) \times \Pi(\mu,\nu)$ is an $\epsilon$-approximate pair of near-optimal subspace projection and optimal transportation plan for the computation of the PRW distance in Eq. (2.3) if the following statements hold true:*

- $\widehat{U}$ *is an $\epsilon$-approximate near-optimal subspace projection of $f$ over $\mathrm{St}(d,k)$ in Eq. (2.7).*

- $\widehat{\pi}$ *is an $\epsilon$-approximate optimal transportation plan for the subspace projection $\widehat{U}$.*

**Algorithm 3** Riemannian SuperGradient Ascent with Network Simplex Iteration (RSGAN)

1: **Input:** measures $\{(x_i, r_i)\}_{i \in [n]}$ and $\{(y_j, c_j)\}_{j \in [n]}$, dimension $k = \widetilde{O}(1)$ and tolerance $\epsilon$.
2: **Initialize:** $U_0 \in \mathrm{St}(d, k)$, $\widehat{\epsilon} \leftarrow \frac{\epsilon}{10\|C\|_\infty}$ and $\gamma_0 \leftarrow \frac{1}{k\|C\|_\infty}$.
3: **for** $t = 0, 1, 2, \ldots, T-1$ **do**
4:     Compute $\pi_{t+1} \leftarrow \mathrm{OT}(\{(x_i, r_i)\}_{i \in [n]}, \{(y_j, c_j)\}_{j \in [n]}, U_t, \widehat{\epsilon})$.
5:     Compute $\xi_{t+1} \leftarrow P_{\mathrm{T}_{U_t}\mathrm{St}}(2V_{\pi_{t+1}} U_t)$.
6:     Compute $\gamma_{t+1} \leftarrow \gamma_0/\sqrt{t+1}$.
7:     Compute $U_{t+1} \leftarrow \mathrm{Retr}_{U_t}(\gamma_{t+1}\xi_{t+1})$.
8: **end for**

---

Finally, we prove that the stationary measure in Definition E.2 is a local surrogate for the stationary measure in Definition 2.7 in the following proposition.

**Proposition E.1** *If* $(U, \pi) \in \mathrm{St}(d, k) \times \Pi(\mu, \nu)$ *is an $\epsilon$-approximate pair of optimal subspace projection and optimal transportation plan of problem* (2.3)*, it is an $3\epsilon$-approximate pair of optimal subspace projection and optimal transportation plan.*

*Proof.* By the definition, $(U, \pi) \in \mathrm{St}(d, k) \times \Pi(\mu, \nu)$ satisfies that $\pi$ is an $\epsilon$-approximate optimal transportation plan for the subspace projection $U$. Thus, it suffices to show that $\Theta(U) \leq 3\epsilon$. By the definition of $p(U)$, we have

$$f(p(U)) - 6\|C\|_\infty \|p(U) - U\|_F^2 \geq f(U).$$

Since $f$ is $2\|C\|_\infty$-weakly concave and each element of the subdifferential $\partial f(U)$ is bounded by $2\|C\|_\infty$ for all $U \in \mathrm{St}(d, k)$, the Riemannian subgradient inequality [55, Theorem 1] implies that

$$f(\mathrm{prox}_{\rho f}(U)) - f(U) \leq \langle \xi, \mathrm{prox}_{\rho f}(U) - U \rangle + 2\|C\|_\infty \|\mathrm{prox}_{\rho f}(U) - U\|^2 \quad \text{for any } \xi \in \mathrm{subdiff}\, f(U).$$

Since $\mathrm{dist}(0, \mathrm{subdiff}\, f(U)) \leq \epsilon$, we have

$$f(\mathrm{prox}_{\rho f}(U)) - f(U) \leq \epsilon\|\mathrm{prox}_{\rho f}(U) - U\|_F + 2\|C\|_\infty \|\mathrm{prox}_{\rho f}(U) - U\|^2.$$

Putting these pieces together with the definition of $\Theta(U)$ yields the desired result. $\qquad\square$

## F    Riemannian Supergradient meets Network Simplex Iteration

In this section, we propose a new algorithm, named *Riemannian SuperGradient Ascent with Network simplex iteration* (RSGAN), for computing the PRW distance in Eq. (2.3). The iterates are guaranteed to converge to an $\epsilon$-approximate pair of *near-optimal* subspace projection and optimal transportation plan (cf. Definition E.2). The complexity bound is $\widetilde{O}(n^2(d+n)\epsilon^{-4})$ if $k = \widetilde{O}(1)$.

### F.1    Algorithmic scheme

We start with a brief overview of the Riemannian supergradient ascent algorithm for nonsmooth Stiefel optimization. Letting $F : \mathbb{R}^{d \times k} \to \mathbb{R}$ be a nonsmooth but weakly concave function, we consider

$$\max_{U \in \mathrm{St}(d,k)} F(U).$$

A generic Riemannian supergradient ascent algorithm for solving this problem is given by

$$U_{t+1} \leftarrow \mathrm{Retr}_{U_t}(\gamma_{t+1}\xi_{t+1}) \quad \text{for any } \xi_{t+1} \in \mathrm{subdiff}\, F(U_t),$$

where $\mathrm{subdiff}\, F(U_t)$ is Riemannian subdifferential of $F$ at $U_t$ and Retr is any retraction on $\mathrm{St}(d, k)$. For the nonconvex nonsmooth optimization, the stepsize setting $\gamma_{t+1} = \gamma_0/\sqrt{t+1}$ is widely accepted in both theory and practice [27, 55].

By the definition of Riemannian subdifferential, $\xi_t$ can be obtained by taking $\xi \in \partial F(U)$ and by setting $\xi_t = P_{\mathrm{T}_U\mathrm{St}}(\xi)$. Thus, it is necessary for us to specify the subdifferential of $f$ in Eq. (2.7). Using the symmetry of $V_\pi$, we have

$$\partial f(U) = \mathrm{Conv}\left\{ 2V_{\pi^\star} U \mid \pi^\star \in \operatorname*{argmin}_{\pi \in \Pi(\mu, \nu)} \langle UU^\top, V_\pi \rangle \right\}, \quad \text{for any } U \in \mathbb{R}^{d \times k}.$$

The remaining step is to solve an OT problem with a given $U$ at each inner loop of the maximization and use the output $\pi(U)$ to obtain an inexact supergradient of $f$. Since the OT problem with a given $U$ is exactly an LP, this is possible and can be done by applying the variant of network simplex method in the POT package [34]. While the simplex method can exactly solve this LP, we adopt the inexact solving rule as a practical matter. More specifically, the output $\pi_{t+1}$ satisfies that $\pi_{t+1} \in \Pi(\mu, \nu)$ and $\|\pi_{t+1} - \pi_t^\star\|_1 \leq \widehat{\epsilon}$ where $\pi_t^\star$ is an optimal solution of unregularized OT problem with $U_t \in \mathrm{St}(d, k)$. With the inexact solving rule, the interior-point method and some first-order methods can be adopted to solve the unregularized OT problem. To this end, we summarize the pseudocode of the RSGAN algorithm in Algorithm 3.

### F.2 Complexity analysis for Algorithm 3

We define a function which is important to the subsequent analysis of Algorithm 3:

$$\Phi(U) := \max_{U' \in \mathrm{St}(d,k)} \left\{ f(U') - 6\|C\|_\infty \|U' - U\|_F^2 \right\} \quad \text{for all } U \in \mathrm{St}(d, k).$$

Our first lemma provides a key inequality for quantifying the progress of the iterates $\{(U^t, \pi^t)\}_{t \geq 1}$ generated by Algorithm 3 using $\Phi(\bullet)$ as the potential function.

**Lemma F.1** *Letting $\{(U_t, \pi_t)\}_{t \geq 1}$ be the iterates generated by Algorithm 3, we have*

$$\begin{aligned}
\Phi(U_{t+1}) \quad \geq \quad & \Phi(U_t) - 12\gamma_{t+1}\|C\|_\infty \left( f(U_t) - f(p(U_t)) + 4\|C\|_\infty \|p(U_t) - U_t\|_F^2 + \frac{\epsilon^2}{200\|C\|_\infty} \right) \\
& - \; 200\gamma_{t+1}^2 \|C\|_\infty^3 (\gamma_{t+1}^2 L_2^2 \|C\|_\infty^2 + \gamma_{t+1}\|C\|_\infty + \sqrt{k}).
\end{aligned}$$

*Proof.* Since $p(U_t) \in \mathrm{St}(d, k)$, we have

$$\Phi(U_{t+1}) \; \geq \; f(p(U_t)) - 6\|C\|_\infty \|p(U_t) - U_{t+1}\|_F^2. \tag{F.1}$$

Using the update formula of $U_{t+1}$, we have

$$\|p(U_t) - U_{t+1}\|_F^2 \; = \; \|p(U_t) - \mathrm{Retr}_{U_t}(\gamma_{t+1}\xi_{t+1})\|_F^2.$$

Using the Cauchy-Schwarz inequality and Proposition A.1, we have

$$\begin{aligned}
& \|p(U_t) - \mathrm{Retr}_{U_t}(\gamma_{t+1}\xi_{t+1})\|_F^2 \\
= \quad & \|(U_t + \gamma_{t+1}\xi_{t+1} - p(U_t)) + (\mathrm{Retr}_{U_t}(\gamma_{t+1}\xi_{t+1}) - U_t - \gamma_{t+1}\xi_{t+1})\|_F^2 \\
\leq \quad & \|U_t + \gamma_{t+1}\xi_{t+1} - p(U_t)\|_F^2 + \|\mathrm{Retr}_{U_t}(\gamma_{t+1}\xi_{t+1}) - (U_t + \gamma_{t+1}\xi_{t+1})\|_F^2 \\
& + 2\|U_t + \gamma_{t+1}\xi_{t+1} - p(U_t)\|_F \|\mathrm{Retr}_{U_t}(\gamma_{t+1}\xi_{t+1}) - (U_t + \gamma_{t+1}\xi_{t+1})\|_F \\
\leq \quad & \|U_t + \gamma_{t+1}\xi_{t+1} - p(U_t)\|_F^2 + \gamma_{t+1}^4 L_2^2 \|\xi_{t+1}\|_F^4 + 2\gamma_{t+1}^2 \|U_t + \gamma_{t+1}\xi_{t+1} - p(U_t)\|_F \|\xi_{t+1}\|_F^2 \\
\leq \quad & \|U_t - p(U_t)\|_F^2 + 2\gamma_{t+1}\langle \xi_{t+1}, U_t - p(U_t)\rangle + \gamma_{t+1}^2 \|\xi_{t+1}\|_F^2 + \gamma_{t+1}^4 L_2^2 \|\xi_{t+1}\|_F^4 \\
& + 2\gamma_{t+1}^2 \|U_t + \gamma_{t+1}\xi_{t+1} - p(U_t)\|_F \|\xi_{t+1}\|_F^2.
\end{aligned}$$

Since $U_t \in \mathrm{St}(d, k)$ and $p(U_t) \in \mathrm{St}(d, k)$, we have $\|U_t\|_F \leq \sqrt{k}$ and $\|p(U_t)\|_F \leq \sqrt{k}$. By the update formula for $\xi_{t+1}$, we have

$$\|\xi_{t+1}\|_F \; = \; \|P_{\mathrm{T}_{U_{t-1}}\mathrm{St}}(2V_{\pi_{t+1}}U_t)\|_F \; \leq \; 2\|V_{\pi_{t+1}}U_t\|_F.$$

Since $U_t \in \mathrm{St}(d, k)$ and $\pi_{t+1} \in \Pi(\mu, \nu)$, we have $\|\xi_{t+1}\|_F \leq 2\|C\|_\infty$. Putting all these pieces together yields that

$$\begin{aligned}
\|p(U_t) - U_{t+1}\|_F^2 \quad \leq \quad & \|U_t - p(U_t)\|_F^2 + 2\gamma_{t+1}\langle \xi_{t+1}, U_t - p(U_t)\rangle + 4\gamma_{t+1}^2\|C\|_\infty^2 \quad \text{(F.2)} \\
& + 16\gamma_{t+1}^4 L_2^2 \|C\|_\infty^4 + 16\gamma_{t+1}^3 \|C\|_\infty^3 + 16\gamma_{t+1}^2 \sqrt{k}\|C\|_\infty^2.
\end{aligned}$$

Plugging Eq. (F.2) into Eq. (F.1) and simplifying the inequality using $k \geq 1$, we have

$$\begin{aligned}
\Phi(U_{t+1}) \quad \geq \quad & f(p(U_t)) - 6\|C\|_\infty \|U_t - p(U_t)\|_F^2 - 12\gamma_{t+1}\|C\|_\infty \langle \xi_{t+1}, U_t - p(U_t)\rangle \\
& - 200\gamma_{t+1}^2 \|C\|_\infty^3 \left( \gamma_{t+1}^2 L_2^2 \|C\|_\infty^2 + \gamma_{t+1}\|C\|_\infty + \sqrt{k} \right).
\end{aligned}$$

By the definition of $\Phi(\bullet)$ and $p(\bullet)$, we have

$$
\begin{aligned}
\Phi(U_{t+1}) \geq{} & \Phi(U_t) - 12\gamma_{t+1}\|C\|_\infty \langle \xi_{t+1}, U_t - p(U_t) \rangle \\
& -200\gamma_{t+1}^2 \|C\|_\infty^3 \left( \gamma_{t+1}^2 L_2^2 \|C\|_\infty^2 + \gamma_{t+1}\|C\|_\infty + \sqrt{k} \right).
\end{aligned} \tag{F.3}
$$

Now we proceed to bound the term $\langle \xi_{t+1}, U_t - p(U_t) \rangle$. Letting $\xi_t^\star = P_{\mathrm{T}_{U_t}\mathrm{St}}(2V_{\pi_t^\star} U_t)$ where $\pi_t^\star$ is a minimizer of unregularized OT problem, i.e., $\pi_t^\star \in \operatorname{argmin}_{\pi \in \Pi(\mu,\nu)} \langle U_t U_t^\top, V_\pi \rangle$, we have

$$
\langle \xi_{t+1}, U_t - p(U_t) \rangle \leq \langle \xi_t^\star, U_t - p(U_t) \rangle + \|\xi_{t+1} - \xi_t^\star\|_F \|U_t - p(U_t)\|_F. \tag{F.4}
$$

Since $f(U) = \min_{\pi \in \Pi(\mu,\nu)} \langle U_t U_t^\top, V_\pi \rangle$ is $2\|C\|_\infty$-weakly concave over $\mathbb{R}^{d \times k}$ (cf. Lemma 2.1), $\xi_t^\star \in \operatorname{subdiff} f(U_t)$ and each element in the subdifferential $\partial f(U)$ is bounded by $2\|C\|_\infty$ for all $U \in \mathrm{St}(d,k)$ (cf. Lemma 2.2), the Riemannian subgradient inequality [55, Theorem 1] holds true and implies that

$$
f(p(U_t)) \leq f(U_t) + \langle \xi_t^\star, p(U_t) - U_t \rangle + 2\|C\|_\infty \|p(U_t) - U_t\|_F^2.
$$

This implies that

$$
\langle \xi_t^\star, U_t - p(U_t) \rangle \leq f(U_t) - f(p(U_t)) + 2\|C\|_\infty \|p(U_t) - U_t\|_F^2. \tag{F.5}
$$

By the definition of $\xi_{t+1}$ and $\xi_t^\star$, we have

$$
\|\xi_{t+1} - \xi_t^\star\|_F = \|P_{\mathrm{T}_{U_t}\mathrm{St}}(2V_{\pi_{t+1}} U_t) - P_{\mathrm{T}_{U_t}\mathrm{St}}(2V_{\pi_t^\star} U_t)\|_F \leq 2\|(V_{\pi_{t+1}} - V_{\pi_t^\star}) U_t\|_F.
$$

By the definition of the subroutine $\mathrm{OT}(\{(x_i, r_i)\}_{i\in[n]}, \{(y_j, c_j)\}_{j\in[n]}, U, \hat{\epsilon})$ in Algorithm 3, we have $\pi_{t+1} \in \Pi(\mu,\nu)$ and $\|\pi_{t+1} - \pi_t^\star\|_1 \leq \hat{\epsilon}$. Thus, we have

$$
\|\xi_{t+1} - \xi_t^\star\|_F \leq 2\|C\|_\infty \hat{\epsilon} \leq \frac{\epsilon}{5}.
$$

Using Young's inequality, we have

$$
\begin{aligned}
\|\xi_{t+1} - \xi_t^\star\|_F \|U_t - p(U_t)\|_F \leq{} & \frac{\|\xi_{t+1} - \xi_t^\star\|_F^2}{8\|C\|_\infty} + 2\|C\|_\infty \|U_t - p(U_t)\|_F^2 \\
\leq{} & \frac{\epsilon^2}{200\|C\|_\infty} + 2\|C\|_\infty \|U_t - p(U_t)\|_F^2.
\end{aligned} \tag{F.6}
$$

Combining Eq. (F.3), Eq. (F.4), Eq. (F.5) and Eq. (F.6) yields the desired result. $\qquad\square$

Putting Lemma F.1 together with the definition of $p(\bullet)$, we have the following consequence:

**Proposition F.2** *Letting $\{(U_t, \pi_t)\}_{t \geq 1}$ be the iterates generated by Algorithm 3, we have*

$$
\begin{aligned}
& \frac{24\|C\|_\infty^2 \sum_{t=0}^{T-1} \gamma_{t+1} \|p(U_t) - U_t\|_F^2}{\sum_{t=0}^{T-1} \gamma_{t+1}} \\
& \leq \frac{\gamma_0^{-1} \Delta_\Phi + 200\gamma_0 \|C\|_\infty^3 (\gamma_0^2 L_2^2 \|C\|_\infty^2 + \gamma_0 \|C\|_\infty + \sqrt{k}(\log(T) + 1))}{2\sqrt{T}} + \frac{\epsilon^2}{12},
\end{aligned}
$$

*where $\Delta_\Phi = \max_{U \in \mathrm{St}(d,k)} \Phi(U) - \Phi(U_0)$ is the initial objective gap.*

*Proof.* By the definition of $p(\bullet)$, we have

$$
\begin{aligned}
& f(U_t) - f(p(U_t)) + 4\|C\|_\infty \|p(U_t) - U_t\|_F^2 \\
={} & f(U_t) - \left( f(p(U_t)) - 6\|C\|_\infty \|p(U_t) - U_t\|_F^2 \right) - 2\|C\|_\infty \|p(U_t) - U_t\|_F^2 \\
\leq{} & -2\|C\|_\infty \|p(U_t) - U_t\|_F^2.
\end{aligned}
$$

Using Lemma F.1, we have

$$
\begin{aligned}
\Phi(U_{t+1}) \geq{} & \Phi(U_t) + 24\gamma_{t+1}\|C\|_\infty^2 \|p(U_t) - U_t\|_F^2 - \frac{\gamma_{t+1}\epsilon^2}{12} \\
& -200\gamma_{t+1}^2 \|C\|_\infty^3 (\gamma_{t+1}^2 L_2^2 \|C\|_\infty^2 + \gamma_{t+1}\|C\|_\infty + \sqrt{k}).
\end{aligned}
$$

Rearranging this inequality, we have

$$
\begin{aligned}
24\gamma_{t+1}\|C\|_\infty^2 \|p(U_t) - U_t\|_F^2 \;\leq\;\; & \Phi(U_{t+1}) - \Phi(U_t) + \frac{\gamma_{t+1}\epsilon^2}{12} \\
& + 200\gamma_{t+1}^2\|C\|_\infty^3(\gamma_{t+1}^2 L_2^2\|C\|_\infty^2 + \gamma_{t+1}\|C\|_\infty + \sqrt{k}).
\end{aligned}
$$

Summing up over $t = 0, 1, 2, \ldots, T-1$ yields that

$$
\frac{24\|C\|_\infty^2 \sum_{t=0}^{T-1} \gamma_{t+1}\|p(U_t) - U_t\|_F^2}{\sum_{t=0}^{T-1} \gamma_{t+1}} \;\leq\; \frac{\Delta\Phi + 200\|C\|_\infty^3 (\sum_{t=1}^{T} \gamma_t^2(\gamma_t^2 L_2^2\|C\|_\infty^2 + \gamma_t\|C\|_\infty + \sqrt{k}))}{2\sum_{t=1}^{T} \gamma_t} + \frac{\epsilon^2}{12}.
$$

By the definition of $\{\gamma_t\}_{t\geq 1}$, we have

$$
\sum_{t=1}^{T} \gamma_t \geq \gamma_0\sqrt{T}, \quad \sum_{t=1}^{T} \gamma_t^2 \leq \gamma_0^2(\log(T) + 1), \quad \sum_{t=1}^{T} \gamma_t^3 \leq 3\gamma_0^3, \quad \sum_{t=1}^{T} \gamma_t^4 \leq 2\gamma_0^4.
$$

Putting these pieces together yields the desired result. $\qquad\square$

We proceed to provide an upper bound for the number of iterations needed to return an $\epsilon$-approximate near-optimal subspace projection $U_t \in \mathrm{St}(d, k)$ satisfying $\Theta(U_t) \leq \epsilon$ in Algorithm 3.

**Theorem F.3** *Letting $\{(U_t, \pi_t)\}_{t\geq 1}$ be the iterates generated by Algorithm 3, the number of iterations required to reach $\Theta(U_t) \leq \epsilon$ satisfies*

$$
t \;=\; \widetilde{O}\left(\frac{k^2\|C\|_\infty^4}{\epsilon^4}\right).
$$

*Proof.* By the definition of $\Theta(\bullet)$ and $p(\bullet)$, we have $\Theta(U_t) = 12\|C\|_\infty\|p(U_t) - U_t\|_F$. Using Proposition F.2, we have

$$
\frac{\sum_{t=0}^{T-1} \gamma_{t+1}(\Theta(U_t))^2}{\sum_{t=0}^{T-1} \gamma_{t+1}} \;\leq\; \frac{3\gamma_0^{-1}\Delta\Phi + 600\gamma_0\|C\|_\infty^3(\gamma_0^2 L_2^2\|C\|_\infty^2 + \gamma_0\|C\|_\infty + \sqrt{k}(\log(T) + 1))}{\sqrt{T}} + \frac{\epsilon^2}{2}.
$$

Furthermore, by the definition $\Phi(\bullet)$, we have

$$
\begin{aligned}
|\Phi(U)| \;\leq\;\; & \max_{U' \in \mathrm{St}(d,k)} |f(U') + 6\|C\|_\infty\|U' - U\|_F^2| \\
\leq\;\; & \max_{U \in \mathrm{St}(d,k)} \max_{U' \in \mathrm{St}(d,k)} |f(U') + 6\|C\|_\infty\|U' - U\|_F^2| \\
\leq\;\; & \max_{U \in \mathrm{St}(d,k)} |f(U)| + 12k\|C\|_\infty.
\end{aligned}
$$

By the definition of $f(\bullet)$, we have $\max_{U \in \mathrm{St}(d,k)} |f(U)| \leq \|C\|_\infty$. Putting these pieces together with $k \geq 1$ implies that $|\Phi(U)| \leq 20k\|C\|_\infty$. By the definition of $\Delta_\Phi$, we conclude that $\Delta_\Phi \leq 40k\|C\|_\infty$. Given that $\gamma_0 = 1/\|C\|_\infty$ and $\Theta(U_t) > \epsilon$ for all $t = 0, 1, \ldots, T-1$, the upper bound $T$ must satisfy

$$
\epsilon^2 \;\leq\; \frac{240k\|C\|_\infty^2 + 1200\|C\|_\infty^2(L_2^2 + \sqrt{k}\log(T) + \sqrt{k} + 1)}{\sqrt{T}}.
$$

This implies the desired result. $\qquad\square$

Equipped with Theorem F.3 and Algorithm 3, we establish the complexity bound of Algorithm 3.

**Theorem F.4** *The RSGAN algorithm (cf. Algorithm 3) returns an $\epsilon$-approximate pair of near-optimal subspace projection and optimal transportation plan of computing the PRW distance in Eq. (2.3) (cf. Definition E.2) in*

$$
\widetilde{O}\left(\frac{n^2(n+d)\|C\|_\infty^4}{\epsilon^4}\right)
$$

*arithmetic operations.*

Figure 7: Fragmented hypercube with $(n, d) = (100, 30)$ (above) and $(n, d) = (250, 30)$ (bottom). Optimal mappings in the Wasserstein space (left), in the SRW space (middle) and the PRW space (right). Geodesics in the PRW space are robust to statistical noise.

*Proof.* First, Theorem F.3 implies that the iteration complexity of Algorithm 3 is

$$\widetilde{O}\left(\frac{k^2 \|C\|_\infty^4}{\epsilon^4}\right). \tag{F.7}$$

This implies that $U_t$ is an $\epsilon$-approximate near-optimal subspace projection of problem (2.7). Furthermore, $\widehat{\epsilon} = \min\{\epsilon, \epsilon^2/144\|C\|_\infty\}$. Since $\pi_{t+1} \leftarrow \mathrm{OT}(\{(x_i, r_i)\}_{i \in [n]}, \{(y_j, c_j)\}_{j \in [n]}, U_t, \widehat{\epsilon})$, we have $\pi_{t+1} \in \Pi(\mu, \nu)$ and $\langle U_t U_t^\top, V_{\pi_{t+1}} - V_{\pi_t^\star}\rangle \leq \widehat{\epsilon} \leq \epsilon$. This implies that $\pi_{t+1}$ is an $\epsilon$-approximate optimal transportation plan for the subspace projection $U_t$. Therefore, we conclude that $(U_t, \pi_{t+1}) \in \mathrm{St}(d, k) \times \Pi(\mu, \nu)$ is an $\epsilon$-*approximate pair of near-optimal subspace projection and optimal transportation plan* of problem (2.3).

The remaining step is to analyze the complexity bound. Note that the most of software packages, e.g., POT [34], implement the OT subroutine using a variant of the network simplex method with a block search pivoting strategy [26, 13]. The best known complexity bound is provided in Tarjan [82] and is $\widetilde{O}(n^3)$. Using the same argument in Theorem 3.1, the number of arithmetic operations at each loop is

$$\widetilde{O}\left(n^2 dk + dk^2 + k^3 + n^3\right). \tag{F.8}$$

Putting Eq. (F.7) and Eq. (F.8) together with $k = \widetilde{O}(1)$ yields the desired result. □

**Remark F.5** *The complexity bound of Algorithm 3 is better than that of Algorithm 1 and 2 in terms of $\epsilon$ and $\|C\|_\infty$. This makes sense since Algorithm 3 only returns an $\epsilon$-approximate pair of near-optimal subspace projection and optimal transportation plan which is weaker than an $\epsilon$-approximate pair of optimal subspace projection and optimal transportation plan. Furthermore, Algorithm 3 implements the network simplex method as the inner loop which might suffer when $n$ is large and yield unstable performance in practice.*

# G   Implementation Details

**Experimental setup.** For the experiments on the MNIST digits, we run the feature extractor pretrained in PyTorch 1.5. All the experiments are implemented in Python 3.7 with Numpy 1.18 on a ThinkPad X1 with an Intel Core i7-10710U (6 cores and 12 threads) and 16GB memory, equipped with Ubuntu 20.04.

**Fragmented hypercube.** We consider the fragmented hypercube which is also used to evaluate the SRW distance [69] and FactoredOT [35]. In particular, we consider $\mu = \mathcal{U}([-1, 1]^d)$ which is

Figure 8: Optimal 2-dimensional projections between "Dunkirk" and "Interstellar" (left) and optimal 2-dimensional projections between "Julius Caesar" and "The Merchant of Venice" (right). Common words of two items are displayed in violet and the 30 most frequent words of each item are displayed.

an uniform distribution over an hypercube and $\nu = T_{\#}\mu$ which is the push-forward of $\mu$ under the map $T(x) = x + 2\text{sign}(x) \odot (\sum_{k=1}^{k^*} e_k)$. Note that $\text{sign}(\cdot)$ is taken element-wise, $k^* \in [d]$ and $(e_1, \ldots, e_d)$ is the canonical basis of $\mathbb{R}^d$. By the definition, $T$ divides $[-1, 1]^d$ into four different hyper-rectangles, as well as serves as a subgradient of convex function. This together with Brenier's theorem (cf. Villani [86, Theorem 2.12]) implies that $T$ is an optimal transport map between $\mu$ and $\nu = T_{\#}\mu$ with $\mathcal{W}_2^2(\mu, \nu) = 4k^*$. Notice that the displacement vector $T(x) - x$ is optimal for any $x \in \mathbb{R}^d$ and always belongs to the $k^*$-dimensional subspace spanned by $\{e_j\}_{j \in [k^*]}$. Putting these pieces together yields that $\mathcal{P}_k^2(\mu, \nu) = 4k^*$ for any $k \geq k^*$.

**Robustness of $\mathcal{P}_k$ to noise.** We consider the Gaussian distribution[6]. In particular, we consider $\mu = \mathcal{N}(0, \Sigma_1)$ and $\nu = \mathcal{N}(0, \Sigma_2)$ where $\Sigma_1, \Sigma_2 \in \mathbb{R}^{d \times d}$ are positive semidefinite matrices of rank $k^*$. This implies that either of the support of $\mu$ and $\nu$ is the $k^*$-dimensional subspace of $\mathbb{R}^d$. Even though the supports of $\mu$ and $\nu$ can be different, their union is included in a $2k^*$-dimensional subspace. Putting these pieces together yields that $\mathcal{P}_k^2(\mu, \nu) = \mathcal{W}_2^2(\mu, \nu)$ for any $k \geq 2k^*$. In our experiment, we set $d = 20$ and sample 100 independent couples of covariance matrices $(\Sigma_1, \Sigma_2)$, where each has independently a Wishart distribution with $k^* = 5$ degrees of freedom. Then we construct the empirical measures $\widehat{\mu}$ and $\widehat{\nu}$ by drawing $n = 100$ points from $\mathcal{N}(0, \Sigma_1)$ and $\mathcal{N}(0, \Sigma_2)$.

# H  Additional Experimental Results

**Fragmented hypercube.** Figure 7 presents the optimal transport plan in the Wasserstein space (left), the optimal transport plan in the SRW space (middle), and the optimal transport plan in the PRW space (right) between $\widehat{\mu}$ and $\widehat{\nu}$. We consider two cases: $n = 100$ and $n = 250$, in our experiment and observe that our results are consistent with Paty and Cuturi [69, Figure 5], showing that both PRW and SRW distances share important properties with the Wasserstein distance.

**Experiments on Movie and Shakespeare operas.** Figure 8 displays the projection of two measures associated with *Dunkirk* versus *Interstellar* (left) and *Julius Caesar* versus *The Merchant of Venice* (right) onto their optimal 2-dimensional projection.

**Experiments on MNIST data.** To further show the versatility of SRW and PRW distances, we extract the features of different MNIST digits using a convolutional neural network (CNN) and compute the scaled SRW and PRW distances between all pairs of MNIST digits. In particular, we use an off-the-shelf PyTorch implementation[7] and pretrain on MNIST with 98.6% classification accuracy

|     | D0 | D1 | D2 | D3 | D4 | D5 | D6 | D7 | D8 | D9 |
|-----|----|----|----|----|----|----|----|----|----|----|
| D0 | 0/0 | 0.97/0.79 | **0.80/0.59** | 1.20/0.92 | 1.23/0.90 | 1.03/0.71 | 0.81/0.59 | 0.86/0.66 | 1.06/0.79 | 1.09/0.81 |
| D1 | 0.97/0.79 | 0/0 | 0.66/0.51 | 0.86/0.72 | 0.68/0.54 | 0.84/0.70 | 0.80/0.66 | **0.58/0.47** | 0.88/0.71 | 0.85/0.72 |
| D2 | 0.80/0.59 | **0.66/0.51** | 0/0 | 0.73/0.54 | 1.08/0.79 | 1.08/0.83 | 0.90/0.70 | 0.70/0.53 | 0.68/0.52 | 1.07/0.81 |
| D3 | 1.20/0.92 | 0.86/0.72 | 0.73/0.54 | 0/0 | 1.20/0.87 | **0.58/0.43** | 1.23/0.91 | 0.72/0.55 | 0.88/0.64 | 0.83/0.65 |
| D4 | 1.23/0.90 | 0.68/0.54 | 1.08/0.79 | 1.20/0.87 | 0/0 | 1.00/0.75 | 0.85/0.62 | 0.79/0.61 | 1.09/0.78 | **0.49/0.38** |
| D5 | 1.03/0.71 | 0.84/0.70 | 1.08/0.83 | **0.58/0.43** | 1.00/0.75 | 0/0 | 0.72/0.51 | 0.91/0.68 | 0.72/0.53 | 0.78/0.59 |
| D6 | 0.81/0.59 | 0.80/0.66 | 0.90/0.70 | 1.23/0.91 | 0.85/0.62 | **0.72/0.51** | 0/0 | 1.11/0.83 | 0.92/0.66 | 1.22/0.83 |
| D7 | 0.86/0.66 | **0.58**/0.47 | 0.70/0.53 | 0.72/0.55 | 0.79/0.61 | 0.91/0.68 | 1.11/0.83 | 0/0 | 1.07/0.78 | 0.62/**0.46** |
| D8 | 1.06/0.79 | 0.88/0.71 | **0.68/0.52** | 0.88/0.64 | 1.09/0.78 | 0.72/0.53 | 0.92/0.66 | 1.07/0.78 | 0/0 | 0.87/0.63 |
| D9 | 1.09/0.81 | 0.85/0.72 | 1.07/0.81 | 0.83/0.65 | **0.49/0.38** | 0.78/0.59 | 1.22/0.83 | 0.62/0.46 | 0.87/0.63 | 0/0 |

Table 3: Each entry is scaled $\mathcal{S}_k^2/\mathcal{P}_k^2$ distance between different hand-written digits.

on the test set. We extract the 128-dimensional features of each digit from the penultimate layer of the CNN. Since the MNIST test set contains 1000 images per digit, each digit is associated with a measure over $\mathbb{R}^{128000}$. Then we compute the optimal 2-dimensional projection distance of measures associated with each pair of two digital classes and divide each distance by 1000; see Table 3 for the details. The minimum SRW and PRW distances in each row is highlighted to indicate its most similar digital class of that row, which coincides with our intuitions. For example, D1 is sometimes confused with D7 (0.58/0.47), while D4 is often confused with D9 (0.49/0.38) in scribbles.

## Footnotes

[5]The proximal mapping $p(U)$ must exist since the Stiefel manifold is compact, yet may not be uniquely defined. However, this does not matter since $p(U)$ only appears in the analysis for the purpose of defining the surrogate stationarity measure; see Li et al. [55].

[6] Paty and Cuturi [69] conducted this experiment with their projected supergradient ascent algorithm (cf. Paty and Cuturi [69, Algorithm 1]) with the EMD solver from the POT software package. For a fair comparison, we use Riemannian supergradient ascent algorithm (cf. Algorithm 3) with the EMD solver here; see Appendix for the details.

[7] https://github.com/pytorch/examples/blob/master/mnist/main.py