[Reviews · NeurIPS 2020]

Review 1

Summary and Contributions: The Wasserstein distance emerges from the optimal transport (OT) problem and is a powerful metric to compare two probability measures, since it offers nice theoretical properties and relevant practical implications. However, it has major limitations when applied in large-scale settings: since the Wasserstein distance is defined as the solution of a linear program, its computation becomes rapidly excessive as the dimension of the ambient data space increases ; besides, its sample complexity can grow exponentially in the problem dimension. These unfavorable properties have motivated the development of "computational OT" methods in recent years, which define alternative to the Wasserstein distance with better computational and/or statistical properties, and therefore allow the use of OT in machine learning applications. One approach that was recently proposed and has become increasingly popular, consists in computing the Wasserstein distance between lower-dimensional representations for the two distributions to compare. Specifically, the Projection Robust Wasserstein (PRW) distance (also known as Wasserstein Projection Pursuit) builds the representations by projecting orthogonally the d-dimensional distributions into the k-dimensional subspace (k < d) such that the Wasserstein distance between these k-dimensional reductions is maximized. Previous work reported that, under some assumptions regarding the structure of the high-dimensional distributions (cf. the "spiked transport model"), PRW provides a dimension-free sample complexity ; however, the efficient computation of PRW remains a challenge since the underlying max-min optimization problem is non-convex, and one workaround is to compute Subspace Robust Wasserstein distances instead, which relaxes the problem such that it boils down to convex optimization and returns approximate solutions. This paper introduces a novel methodology to efficiently compute PRW (of order 2, based on the Euclidean distance) between finite discrete probability measures: instead of relying a convex surrogate, the original problem is solved with tools from Riemannian optimization. The contributions are summarized below. - The authors formulate PRW as max-min problem (its original formulation is a max-inf problem) and introduce a regularized version of PRW, which is called the "entropic regularized PRW distance" and obtained by adding an entropy term to the minimization problem in PRW. This regularization implies that the function to maximize over the Stiefel manifold is smooth, thus optimization is easy. - They investigate the properties of the function f to maximize in PRW and show that (i) f is not convex and not smooth (this result motivates the use of regularization), and (ii) its subdifferential is bounded by a constant that only depends on the cost matrix. This property (ii), combined with existing results, allows a characterization of the subdifferential of f restricted on the Stiefel manifold: it is obtained by projecting the subdifferential of f into the tangent space. This characterization is useful to derive the Riemannian gradient used in the algorithms explained below. - They develop two algorithms to efficiently compute the entropic regularized PRW distance: "Riemannian gradient ascent with Sinkhorn" (RGAS) algorithm: the max-min problem is solved by alternatively approximating the solution of the regularized OT problem (min problem) with Sinkhorn's algorithm, and performing Riemannian gradient ascent (as mentionned above, the function to maximize in this setting is smooth, thanks to the regularization). "Riemannian Adaptive Gradient Ascent with Sinkhorn Iteration (RAGAS)" algorithm: same procedure as in Algorithm 1, except that here, the Riemannian gradient ascent is adaptive. - The authors then prove that these algorithms return an approximation of the optimal solution to the original PRW problem (i.e., optimal subspace and optimal transport map) in a finite number of iterations. The exact complexity is derived and is the same for RGAS and RAGAS: it is linear in the dimension d, quadratic in the number of samples and depends on the desired approximation precision (parametrized with \epsilon). They underline that this guarantee is hardly generalized (cf. Remark 3.2). - Finally, they provide an empirical analysis that helps understanding more the performance of RGAS and RAGAS algorithms in practice.

Strengths: - This work is relevant for the NeurIPS community since it develops new methods for computational optimal transport, a research field that has been increasingly used for various machine learning/statistical applications in recent years. - The paper is well written. In particular, I appreciate that a theoretical point of view is developed before introducing the algorithms, as this helps understanding the intuition behind these. - The contributions are novel: this paper shows that the intrinsic max-min problem of PRW can be efficiently solved, which, to the best of my knowledge, has never been done before. Besides, exploiting Riemannian optimization in the context of optimal transport is original, and can foster interesting future applications. - The algorithms proposed in this work have important practical implications: they allow a fast approximation of a relevant distance and seem robust in practice. Besides, they are theoretically grounded, which makes this work useful for practitionners and theoreticians.

Weaknesses: - The experiments conducted in this paper are largely inspired by [1] and compare PRW to the Subspace Robust Wasserstein (SRW) distance. The authors argue that their empirical results show that PRW outperforms the Subspace Robust Wasserstein (SRW) distance [1] "in both statistical and computational aspects" (l.90-91). This conclusion is a bit confusing to me for the following reasons. 1) Figures 1, 2, 3 verify that the behavior of PRW is consistent. Specifically, PRW and SRW seem equivalent regarding their empirical performance: see Figure 1 vs. Figure 2 in [1], Figure 2 vs. Figures 3 and 4 in [1], Figure 3 left vs. Figure 3 right. 2) Figure 4: PRW indeed offers the best performance, but is actually followed very closely by SRW and even yields the same error as SRW for 3 noise levels (out of 7). 3) The experiments on real data (Tables 1 and 2 in the main document, and Table 3 in the supplementary doc.) show that the results given by PRW agree with the intuitive truth (e.g., the most similar movie script to Kill Bill 1 is Kill Bill 2) and are "consistently smaller" than SRW (l.273). Therefore, similar to Figures 1, 2 and 3, these results mainly show that the approximate solution for PRW performs nicely, but doesn't outperform SRW. Besides, the fact that PRW is always smaller than SRW in these experiments, might even suggest that PRW has less discriminative power than SRW. 4) Figure 4 (right) compares the execution times of PRW (approximated with RGAS or RAGAS) and SRW. We observe that PRW is faster than SRW, especially when the dimension is high. This indeed illustrates that PRW is more computationally efficient than SRW. 5) Finally, it is not clear to me which of the empirical results illustrates the statistical advantages of PRW compared with SRW (mentioned in l.90). I am actually wondering whether the efficient computation of PRW proposed in this paper should induce a nice sample complexity, or if the observations in [2] might apply in this setting as well. [1] Subspace Robust Wasserstein Distances. Paty and Cuturi, 2019. [2] Estimation of Wasserstein distances in the Spiked Transport Model. Niles-Weed and Rigollet, 2019. - The hyperparameter tuning (for example, \alpha, \beta, \gamma) and initialization steps (l.2 in Algorithms 1 and 2) are not explained clearly enough. Besides, an analysis of the empirical performance with varying \epsilon values is missing (for example, with the same setting as in Figure 4).

Correctness: The methodology seems correct to me, but I would like a clarification on the following aspects: - The RGAS and RAGAS algorithms uses an accelerated Sinkhorn's algorithm in the sense that the solution to the regularized OT is efficiently approximated (see the parameter \hat{\epsilon}, l.211). Can't we apply the same strategy in the Frank-Wolfe algorithm of SRW (Algorithm 2, [1])? How much would it improve the computational efficiency and impact the quality of the approximate solution? - The curves in Figure 1 are concave and increasing, as outlined in the interpretation of the results (l.235). This was observed in [1] as well, but in this case, this is supported by a theoretical result: see Proposition 3 in [1]. Is it possible to extend this result to PRW? On the other hand, for the same Figure 1, is there an explanation on why RGAS is above RAGAS for k < k*, and then under for k >= k*? - In Figure 4, why does PRW brutally decrease when the noise level is equal to 4, and increase again for higher values? - The submission in its current form doesn't contain enough material to reproduce the experimental results. In particular, the implementation of the RGAS/RAGAS algorithms is not provided. I suggest that the authors improve the reproductibility of their results as a way to ensure their correctness, for example by releasing the source code.

Clarity: Overall, the paper is clear. Here is a list a some inaccuracies and typos that I noticed. L.117: why is this mapping lower semi-continuous? L.127: the notations for the marginals r(\pi) and c(\pi) should be introduced in the main document Algorithms 1 and 2: what are L_1 and L_2 in the definition of \gamma? L.175: for clarity, the notion of exponential mapping and its computational challenges could be more developed (in the supplementary document, or by giving references) L.182: "We now present a novel approach to exploiting" L.193: T_{U}St instead of T_{Z}St L.195: the "dist" operator should be explained in the main document L.203: "f_\eta" should be "f_\eta(U)" L.204: "and define the mapping \pi \mapsto ..." what's the use of introducing this mapping here? L.214: precise here that "grad" denotes the Riemannian gradient (gradient of the function on the Stiefel manifold) L.215: "see Algorithm 1" it should be Algorithm 2 L.216: "in terms of robustness" what does the term robustness mean in this context? L.233: "k*" should be defined here Figure 2: define \hat{\Omega} in the legend for y-axis. L.577: "differentiablity"

Relation to Prior Work: The relation to prior work is clearly discussed in the introduction. For future work, it would be nice to compare PRW against more strategies, including the max. sliced Wasserstein distance.

Reproducibility: No

Additional Feedback: *** Edit after response *** I thank the authors for addressing my concerns on the experiments and I appreciate they found my comments helpful. I am still not convinced that the advantageous sample complexity of PRW is "partially demonstrated" by Figure 4 (to me, this is a strong statement, especially since the experiment is run for only one value of number of samples). I also found their answer to my question on the behavior of RGAS and RAGAS in Figure 4 too vague. I am keeping my positive score, but I encourage the authors to interpret more carefully the empirical results in the revised version. *** This work introduces an interesting and original approach to efficiently compute the Projection Robust Wasserstein. However, I think the conclusions of the empirical analysis should be updated: given my comments in the "Weaknesses" section, I think it would be fairer to only say that PRW provides a computational advantage over SRW (provided that the value of \epsilon is not too small).


Review 2

Summary and Contributions: This paper proposes a new approach to compute the projection robust Wasserstein distance between two discrete distributions. The outer maximization problem is solved to first-order stationarity by gradient descent on the Stiefel manifold, whose approximate Riemannian gradient is obtained approximately by solving an entropic regularization OT. Finite-time convergence guarantees and thorough numeircal experiments are provided.

Strengths: (1) Computing the projected Wasserstein distance is an important topic in OT and ML. While previous work focuses on the statistical properties and convex relaxation of the projection robust Wasserstein distance, this paper considers a direct approach by solving a max-min formulation. (2) A careful theoretical convergence guarantee is provided. The methodology is a non-tirival combination of standard techniques from OT and Riemannian optimization. (3) Numerical experiements have been conducted on a variety of instances.

Weaknesses: (1) It should be clearly stated in the introduction that the proposed algorithm has a provable guarantee to a stationary point. The way it is written now seems a bit vague (eps-approximate, efficient algorithm for the computation, etc). (2) Theorem 3.1 provides a convergence guarantee to stationary points. Nevertheless, it would be helpful to provide some insights on how such a local optimal solution is compared to the global optimal solution of a convex relaxation problem studied in [64]. (3) Numerical results are helpful to get insights for (2) especially the results on MNIST. But given that the intrinsic dimension of MNIST is very low, it would be helpful to see the comparison on other datasets with a higher intrinsic dimension. == Edit after rebuttal == I am satisfied with the authors' response.

Correctness: Correct to the best of my knowledge.

Clarity: Yes. It is relatively easy to follow for readers within the field but may contain jargon for the general audience.

Relation to Prior Work: Yes.

Reproducibility: Yes

Additional Feedback:


Review 3

Summary and Contributions: This work propose a max-min optimization model for computing the PRW distance over the Stiefel manifold, and propose Riemannian gradient ascent with Sinkhorn (RGAS) and Riemannian adaptive gradient ascent with Sinkhorn (RAGAS) for entropic regularized PRW distance with theoretical guarantee.

Strengths: WPP is a new variant of the Wasserstein distance to solve the curse of dimensionality, and this paper propose applying Riemmanian technique method to solve WPP. They also prove the convergence result for algorithm. They provide comprehensive empirical studies to evaluate our algorithms on synthetic and real datasets.

Weaknesses: I admit this work has relevance to optimal transport in terms of numerical optimisation method, and it is interesting to exploit more efficient technique for optimising WPP. However, the main problem is this work only focus on Riemannian optimatzation part. So the idea is not very attractive to me. Instead I'd like to see more statistical properties or theoretical analysis about the WPP distance except existing results in ref [64]. For instance, (25) introduce the entropic regularized to PRW distance, indeed, this will facilitate the optimisation. But the question is the statistical part is weak. What is the statistical property of (25)? e.g. whether (25) is strong consistent with original PRW or Wasserstein distance? Is (25) still the well-defined distance on Wasserstein manifold as PRW? There is no such discussion or analysis in this work.

Correctness: yes

Clarity: yes

Relation to Prior Work: Yes

Reproducibility: Yes

Additional Feedback: The response from the authors partially address my concerns. So I raise my socre.


Review 4

Summary and Contributions: The paper concerns computational tools for the projection robust Wasserstein distance / Wasserstein projection pursuit. Some definitions are given for how the non-convex, non-smooth objective is solved in a meaningful (epsilon-approximate) way. Then, solvers based on entropic regularization are presented, for which theoretical guarantees of finite running times are given. The solver relies on the standard Sinkhorn-Knopp algorithm and Riemannian gradient ascent on the Stiefel manifold. Two varients, adaptive and non-adaptive versions, RAGAS and RGAS are provided, which are then demonstrated in the experimental section, and also compared to the subspace robust wasserstein distance (SRW).

Strengths: The substance is novel and clearly of importance to the field of computational OT, which is well motivated by the text, as tools to battle the curse of dimensionality in OT are needed. Is is a natural development for the prior work, by showing that the PRW can be quite efficiently computed in practice. Furthermore, the experimental section is able to demonstrate the efficiency of the method(s).

Weaknesses: My only objection to this paper would be the amount of substance it contains, making me wonder if this conference is the right venue for it. As it is, the paper is crammed full, and comes with a supplementary material of 30 pages, containing ~10 pages of results + their proofs, where the rest consists mostly of additional details on more well-known material to help the reader. On top of this, the experimental section of the paper is not self sufficient, and requires reading the supplementary material in order to understand what has been carried out, e.g., k^* is not defined, and the experimental settings are not explained, except for the real data experiment. The situation could be improved by attempting to make the experimental section more self-contained.

Correctness: The limited review time prevents me from going through in detail the results in the 30 page supplementary material, which will lower my confidence score.

Clarity: The paper is written in a nice way and contains plenty of technical details written in an understandable way.

Relation to Prior Work: The paper explains well the weaknesses of OT that PRW/WPP try to remedy, and builds naturally on these methods.

Reproducibility: Yes

Additional Feedback: ## Edit after rebuttal After reading the other reviews and the rebuttal, I retain the previous score and thank the authors for addressing my concern.

[Author Response · NeurIPS 2020]

**We would like to thank the reviewers for their efforts on reading and evaluating our paper.** We appreciate that they pointed out the importance of the problem and of our analysis given the increasing popularity of computational OT. All the minor comments will be addressed in the paper draft, and we will release the source code if the paper is accepted. In what follows, we provide specific responses to each reviewer.

**Reviewer 1.** We thank R1 for a very detailed evaluation and truly helpful feedback on our empirical analysis. We will use the sentence "PRW provides a computational advantage over SRW (provided that $\epsilon$ is not too small)" as you suggest to conclude our empirical analysis in the updated version.

◆ *Comparison between PRW and SRW:* We present Figures 1-3 and Tables 1-3 to show that it is reasonable to compute the PRW distance by our algorithm. To be more specific, while the SRW distance can be globally optimized, our algorithms only return an approximate stationary point which needs to be evaluated in practice. We agree that SRW has more discriminative power than PRW since it is equivalent to the Wasserstein distance [Prop.2, 64]. However, it also suffers from the curse of dimensionality in theory despite of practical performance. Finally, we highlight that the ideal sample complexity of PRW is partially demonstrated by its robustness to input data. As shown by Figure 4, PRW is more robust than SRW and Wasserstein distance, when the noise has the moderate to high variance.

◆ *The hyperparameter tuning [...] and initialization steps [...] are not explained clearly enough* We will move necessary details of hyperparameter tuning in Appendix H to the main paper to make the experiments clear. Furthermore, we will add the experiments with varying $\epsilon$ values with the same setting as in Figure 4 in the updated version.

◆ *Can't we apply the same strategy in the Frank-Wolfe algorithm of SRW [Alg.2, 64]:* Yes, we can apply the same strategy as in [Alg.2, 64] and it improves the computational efficiency and affects the quality of the approximate solution. Indeed, we made good use of the open source code they have provided to inspire our implementation.

◆ *Is it possible to extend this result to PRW?* We demonstrate that it is difficult to extend Prop. 3 from [64] to PRW. Indeed, the characterization of SRW as a sum of eigenvalues is crucial to the analysis but PRW does not have such property. Furthermore, as $k$ becomes larger, the nonconvex max-min optimization problem has more stationary points. It is possible that RGAS and RAGAS converge to different stationary points with different PRW values.

◆ *is there an explanation on why RGAS is above RAGAS for $k < k^*$, and then under for $k \geq k^*$?* We provide one possible reason for the behavior of PRW value in Figure 4. Indeed, when $\sigma = 4$, our algorithm can converge to a stationary point where the PRW value is closer to the PRW value with $\sigma = 0$.

**Reviewer 2.** We thank R2 for the positive evaluation and very helpful feedback on the theoretical part.

◆ *Convergence to a stationary point:* We will fix this confusing point in the updated version.

◆ *Comparing a local optimal solution to the global optimal solution of a convex relaxation problem in [64]:* Investigating this relationship is difficult in general due to multiple stationary points of non-convex max-min problem but seems possible if data has certain structure. We will add a remark to elaborate our insights in the updated version.

◆ *High-dimensional data:* We agree and will present some results on high-dimensional text in the updated version.

**Review 3.** We thank R3 for an informative and thought provoking review.

◆ *I admit this work has relevance to optimal transport in terms of numerical optimisation method, and it is interesting to exploit more efficient technique for optimising WPP.* Indeed, we have observed in the field of computational OT that the discovery of efficient computational approaches often precede other types of advances (applied or statistical)

◆ *However, the main problem is this work only focus on Riemannian optimatzation part. So the idea is not very attractive to me.* Although we believe the statistical properties of entropic regularized WPP/PRW distance are definitively worth investigating, *we do not understand why our focus here on computational aspects should appear to you as a problem.* These are two distinct and complementary subjects. We argue that studying computational aspects for WPP/PRW distance with theoretical guarantee is needed for these tools to take off. Our methodology is new, and uses a nontrivial combination of techniques from OT and Riemannian optimization. Luckily, we are aware of recent work [https://arxiv.org/abs/2006.12301] that might answer some of your questions. Taken together, these two contributions provide new directions for computational OT.

**Reviewer 4.** We thank R4 for a positive evaluation and very helpful feedback on the paper organization.

◆ *The experimental section is not self sufficient:* As suggested, we will move some parts of implementation details back to the main context and make the experimental section self-contained.

[Meta-Review · NeurIPS 2020]

All reviewers liked the paper and were satisfied with the rebuttal. This paper has made a clear theoretical advancement.